# Matrix games between full siblings in Mendelian populations

**József Garay[1], Tamás Varga** **[2,3]\*, Villő Csiszár[4], Tamás F. Móri[5], András Szilágyi[1]**

**1** HUN-REN Centre for Ecological Research, Institute of Evolution, Budapest, Hungary, **2** Bolyai Institute, University of Szeged, Szeged, Hungary, **3** National Laboratory for Health Security, University of Szeged, Szeged, Hungary, **4** Department of Probability Theory and Statistics, Eötvös Loránd University, Budapest, Hungary, **5** HUN-REN Alfréd Rényi Institute of Mathematics, Budapest, Hungary

\* vargata@math.u-szeged.hu

## Abstract

We develop a model that integrates evolutionary matrix game theory with Mendelian genetics. Within this framework, we define the genotype dynamics that describes how the frequencies of genotypes change in sexual diploid populations. We show that our formal definition of evolutionary stability for genotype distributions implies the stability of the corresponding interior equilibrium point in the genotype dynamics. We apply our findings to a model of familial selection, where the survival rates of siblings in monogamous families are determined by a matrix game between them. According to Mendelian inheritance, the behaviour associated with each genotype is uniquely determined by an autosomal (recessive-dominant or intermediate) allele pair. We provide general conditions for the evolutionary stability of homozygote populations. We find that the payoff matrix and the genotype-phenotype map together determine this stability. In numerical examples we consider the prisoner's dilemma between siblings. Based on the evolutionary stability of the pure cooperator and defector states, we provide a potential classification of the genotype dynamics. We distinguish between two cases: one in which the total survival rate is higher in cooperator-cooperator interactions ("coordinated" case), and another in which it is higher in cooperator-defector interactions ("anti-coordinated" case). In the coordinated case, global stability of cooperator homozygote population is possible but not necessary, since bistability, stable coexistence of cooperators and defectors, and even global stability of the defector homozygote state are all possible, depending on the interaction between the phenotypic payoff matrix and the genotype-phenotype mapping. In the anti-coordinated case, the cooperator homozygote population cannot be stable. Thus, similarly to the group selection theory, the welfare of the family (the sum of the survival rates of siblings) governs the emergence of cooperative behavior among family members. Finally, in the case of the donation game, the classical Hamilton's rule determines whether the homozygous cooperator or the homozygous defector population is stable; bistability or stable coexistence are impossible.

**Data availability statement:** All relevant data are within the manuscript and in Supporting Information files.

**Funding:** This research was supported by the Hungarian National Research, Development and Innovation Office NKFIH (grant number 125569 received by TFM, grant number KKP 129848 received by ASz). This research was supported by NKFIH, Hungary KKP 129877 received by TV. This research was supported by the Bolyai János Research Fellowship of the Hungarian Academy of Sciences received by ASz. The study was funded by the National Research, Development and Innovation Office in Hungary (RRF-2.3.1-21-2022-00006) (to TV). This work was supported by project TKP2021-NVA-09 (to TV). Project no TKP2021-NVA-09 has been implemented with the support provided by the Ministry of Culture and Innovation of Hungary from the National Research, Development and Innovation Fund, financed under the TKP2021-NVA funding scheme. The funders had no role in study design, data collection and analysis, decision to publish, or preparation of the manuscript.

**Competing interests:** The authors have declared that no competing interests exist.

## Introduction

Our motivation stems from a century-old problem raised by Haldane, who coined the term "*familial selection*" [1]. In this scenario, family size is strictly constrained by limited resources, such as food, resulting in the production of more offspring than can survive. This leads to intense competition among siblings [2]. In our study, we investigate a slightly different setting in which survival within monogamous and exogamous families is governed by a frequency-dependent matrix game played between siblings [3,4].

Most models examining interactions between siblings assume a sexually reproducing haploid population (or at least focus primarily on allele frequencies) [5–7], or they consider asexual reproduction, where relatedness is set to 1/2 and this is interpreted as interactions between siblings [8–16]. The latter approach originates with Hamilton [17], who proposed the idea that individuals in large populations do not interact entirely at random but instead are more likely to interact with others who share genes identical by descent [18,19]. In contrast, this paper focuses on diploid organisms with sexual reproduction, where siblings within a family inherit their genes from the same mother and father, according to Mendelian inheritance rules. This naturally ensures a relatedness of 1/2 between full siblings. Furthermore, we explicitly track the frequencies of diploid genotypes across generations.

Sexual reproduction involves alternating haploid and diploid phases, and accordingly, two main modelling approaches are used in evolutionary biology: tracking either the haploid stage (gamete types) or the diploid stage (genotypes) across generations.

*Gene-centred models* assume random mating of gametes. The diploid parental population is taken to be in Hardy–Weinberg equilibrium, and the focus is on allele frequencies [20–22].

*Genotype-centred models* become necessary when internal fertilization occurs and the survival rate of the juveniles depends on the specific mating pair. These models track the frequencies of diploid genotypes directly [3,23–34]. Although diploid embryos follow Hardy-Weinberg proportions under panmixia, family-dependent survival selection can substantially alter the genotype proportions in the adult population. As a result, the Hardy-Weinberg equilibrium becomes inadequate for both the formulation and the analysis of such models [22].

Since the model discussed in this article is genotype-centred, we highlight two further benefits of this approach. First, the classical Darwinian fitness is defined as the growth rate associated with each phenotype—that is, the rate at which individuals of a given phenotype reproduce. Therefore, genotype-centred models align closely with the classical Darwinian perspective, as the genotype determines the phenotype. Second, kin selection theory focuses on diploid individuals, considering the costs and benefits arising from their interactions, and the genetic relatedness between them [20]. In this respect, both classical Hamilton's rule and the notion of inclusive fitness are also genotype-centred.

Genotype-centred population genetic models have been widely used to investigate the evolution of altruism within families [3,4,23,27,29,30,35]. Here, we briefly recall

some key insights from these models. First, the fixation of altruistic behaviour critically depends on the genotype-phenotype mapping [3,29,36]. Second, in the case of additive cost and benefit functions, population genetic models reproduce the same results as Hamilton's method [3,4,17]. However, this equivalence breaks down under non-additive situations [4,23].

Most of the aforementioned models focus on the stability of the altruistic homozygote. In contrast, our interest lies in the evolutionary stability of polymorphic diploid populations—specifically, in the potential coexistence of different genotypes. Studying the stability of such mixed genotype distributions requires a dynamical model. When diploid genotypes mate randomly according to a mating table, we have previously introduced a corresponding genotype dynamics [3,37]. Mating-table-based models have in fact already been applied to evolutionary questions involving maternal-foetal incompatibility (e.g., [25,38]). Using the genotype dynamics, we address the following two questions: 1. What are the static conditions for the existence of a stable mixed genotype distribution? 2. Do these conditions imply the dynamic stability of the corresponding equilibrium in the genotype dynamics [3]?

Furthermore, altruism can be viewed as a one player game, in which an individual chooses between two pure strategies (altruistic or selfish), and the recipient plays no strategic role. This raises the question: does the classical Hamilton's rule still apply if the recipient also has a strategy, that is, if the interaction between siblings is modelled as a two player matrix game?

As an application, we consider the case of two alleles at an autosomal locus, which uniquely determine phenotypes under dominant-recessive or intermediate inheritance. Our goal is to understand how the payoff matrix and the genotype-phenotype mapping jointly influence the long-term outcome of natural selection. To this end, we present numerical examples in which sibling interactions are modelled using the prisoner's dilemma [39] and one of its special cases, the donation game [40].

## Results

### Genotype dynamics in sexual diploid population

We consider a very large diploid population consisting of $N$ individuals. Here by "very large", we mean that during sexual reproduction all possible genotypes occur in the population. Females and males differ only in their sex, the sex ratio is 1:1, and there is no sexual selection. We track the evolution of the genotype frequencies over time. Let $m$ denote the number of distinct genotypes; then, the population state at any given time is represented by a point in the $m$-dimensional simplex

$$S_m = \left\{ x = (x_1, x_2, \ldots, x_m) \in R^m : 0 \leq x_i \leq 1, i = 1, 2, \ldots, m, \text{ and } \sum x_i = 1 \right\},$$

where $x_i$ denotes the frequency of the $i$-th phenotype. Our dynamics is based on the following assumptions.

Assumption A) *The mating system is fixed*. Let the probability of mating of genotypes $G_i$ and $G_j$ be $h_{ij}(x) \in [0, 1]$, e.g., $h_{ij}(x) = x_i x_j$ under panmixia.

Assumption B) *The genotypes of the parents determine the genotypes of their offspring*.

The values $p_{k(ij)} \in [0, 1]$, representing the probability that a mating between genotypes $G_i$ and $G_j$ produces an offspring of genotype $G_k$, are organized into a mating table (Table 1).

**Remark 1**. This framework allows us to consider arbitrary genetic architectures. For instance, the probabilities $p_{k(ij)}$ can be derived for systems with multiple autosomal loci located on the same or different chromosomes, with or without recombination. Furthermore, if each allele can mutate to any other allele at the same locus, then each entry of the mating table is strictly positive.

Since natural selection acts at the level of the phenotype, it is essential to specify the genotype-phenotype map.

Assumption C) *The phenotype is genetically determined*. Under this assumption, we can consider various selection regimes and calculate the survival rate of each sibling based on game-theoretical conflicts (see Remark 2). This condition ensures that the choice of a strategy is not an individual decision but is entirely determined by genotype.

 

**Table 1. Genotype survival table based on the mating table for a general matrix game.**

| Genotypes of parents | Average number of couples assuming panmixia | Average number of surviving offspring, for genotypes $G_1$, $G_2$, $G_3$ | | |
|---|---|---|---|---|
| | | $G_1 = ([a],[a])$ <br> $s_1$ | $G_2 = ([a],[A])$ <br> $s_2$ | $G_3 = ([A],[A])$, <br> $s_3$ |
| $G_1 \times G_1$ | $\frac{N}{2}x_1^2$ | $p_{1(11)} = 1$ <br> $n_{1(11)} = n s_1 A s_1$ | $p_{2(11)} = 0$ <br> thus <br> $n_{2(11)} = 0$ | $p_{3(11)} = 0$ <br> thus <br> $n_{3(11)} = 0$ |
| $G_1 \times G_2$ | $N x_1 x_2$ | $p_{1(12)} = \frac{1}{2}$ <br> $n_{1(12)} = \frac{n}{4}(s_1 A s_1 + s_1 A s_2)$ | $p_{2(12)} = \frac{1}{2}$ <br> $n_{2(12)} = \frac{n}{4}(s_2 A s_1 + s_2 A s_2)$ | $p_{3(12)} = 0$ <br> thus <br> $n_{3(12)} = 0$ |
| $G_1 \times G_3$ | $N x_1 x_3$ | $p_{1(13)} = 0$ <br> thus <br> $n_{1(13)} = 0$ | $p_{2(13)} = 1$ <br> $n_{2(13)} = n s_2 A s_2$ | $p_{3(13)} = 0$ <br> thus <br> $n_{3(13)} = 0$ |
| $G_2 \times G_2$ | $\frac{N}{2}x_2^2$ | $p_{1(22)} = \frac{1}{4}$ <br> $n_{1(22)} =$ <br> $\frac{n}{16}(s_1 A s_1 + 2 s_1 A s_2 + s_1 A s_3)$ | $p_{2(22)} = \frac{1}{2}$ <br> $n_{2(22)} =$ <br> $\frac{n}{8}(s_2 A s_1 + 2 s_2 A s_2 + s_2 A s_3)$ | $p_{3(22)} = \frac{1}{4}$ <br> $n_{3(22)} =$ <br> $\frac{n}{16}(s_3 A s_1 + 2 s_3 A s_2 + s_3 A s_3)$ |
| $G_2 \times G_3$ | $N x_2 x_3$ | $p_{1(23)} = 0$ <br> thus <br> $n_{1(23)} = 0$ | $p_{2(23)} = \frac{1}{2}$ <br> $n_{2(23)} = \frac{n}{4}(s_2 A s_2 + s_2 A s_3)$ | $p_{2(23)} = \frac{1}{2}$ <br> $n_{3(23)} = \frac{n}{4}(s_3 A s_2 + s_3 A s_3)$ |
| $G_3 \times G_3$ | $\frac{N}{2}x_3^2$ | $p_{1(33)} = 0$ <br> thus <br> $n_{1(33)} = 0$ | $p_{1(33)} = 0$ <br> thus <br> $n_{2(33)} = 0$ | $p_{1(33)} = 1$ <br> $n_{3(33)} = n s_2 A s_2$ |

[a] and [A] denote different alleles of the same gene. The phenotypes of corresponding to genotypes $G_1$, $G_2$ and $G_3$ are denoted by $s_1$, $s_2$ and $s_3$, respectively. $x_i$ is the frequency of the genotype $G_i$ in the parental population. $p_{k(ij)}$ is the probability that an offspring has genotype $G_k$, given that the parental genotypes are $G_i$ and $G_j$, respectively. $n_{k(ij)}$ is the number of surviving offspring with genotype $G_k$ provided the parental genotypes are $G_i$ and $G_j$, respectively. $A = (a_{ij})_{2 \times 2}$ denotes the payoff matrix describing the interactions between siblings. $N$ is the size of the parental population. $n$ denotes the total number of newborn offspring per family, which is assumed to be fixed.

Assumption D) *The survival rate of each offspring depends on the genotypes of the parents.* The parents' genotypes determine the genotypes of their offspring, which in turn determine their phenotypes and hence their survival. Assumptions C and D allow us to model survival games between siblings, where the outcome is determined by the parental genotypes. Let $n_{k(ij)}$ denote the number of surviving offspring of genotype $G_k$ resulting from a mating between genotypes $G_i$ and $G_j$.

**Remark 2.** The parameters $p_{k(ij)}$ and $n_{k(ij)}$ are general in the sense that they can accommodate a wide range of selection situations. For instance, the number of offspring may depend on the genotypes of the parents [20] as well as on the genotype distribution $x$. Similarly, the survival probability of an offspring with genotype $G_k$ born to parents with genotypes $G_i$ and $G_j$ may also depend on $x$.

Let $o_{(ij)}(x)$ denote the number of offspring produced by a mating pair with genotypes $G_i$ and $G_j$, and let $\rho_{k(ij)}(x) \in [0,1]$ be the survival probability of an offspring with genotype $G_k$ from such a mating pair, in a population with genotype distribution $x$. Then the number of surviving offspring of genotype $G_k$ is given by

$$n_{k(ij)}(x) = \rho_{k(ij)}(x) p_{k(ij)}(x) o_{(ij)}(x).$$

Nevertheless, after presenting the general formulation of the genotype dynamics, for the remainder of the paper we assume $p_{k(ij)}(x)$ and $n_{k(ij)}(x)$ are independent of $x$ and hence we write them simply as $p_{k(ij)}$ and $n_{k(ij)}$.

To follow the evolution of genotype frequencies over time, we need to know the relative frequency of mating pairs $G_i \times G_j$, denoted by $h_{ij}(x)$, and, for each such pair, the number of surviving offspring with genotype $G_k$, i.e., $n_{k(ij)}(x)$, for any $x \in S_m$.

The total number of offspring of genotype $G_k$ produced by the entire parental population in state $x$ is given by:

$$V_k(x) = \frac{N}{2} \sum_{i,j} h_{ij}(x) n_{k(ij)}(x).$$

If $x \in \text{int} S_m$, the *average production rate* of genotype $G_i$ is

$$U_i(x) = \frac{V_i(x)}{x_i},$$

and the average production rate of the entire diploid population is

$$\overline{U}(x) = \sum_i V_i(x) = \sum_i x_i U_i(x).$$

To avoid degenerate cases, we assume $\overline{U}(x) > 0$ for all $x$, that is, for every $x$, there exists at least one $i$ such that $V_i(x) > 0$.

**Genotype dynamics**: Following the reasoning that precedes the introduction of the replicator dynamics in Chapter 7.1 of [41], we consider a sufficiently large population in which generations blend continuously into each other. In this setting, the frequency of a given genotype can be treated as a differentiable function of time, and its change can be described by a system of differential equations. Analogous to the replicator dynamics, the rate of change is given by the difference between the production rate of the genotype and the average production rate of the population:

$$\dot{x}_i = V_i(x) - x_i \sum_j V_j(x), \qquad i \in \{1, 2, \ldots, m\}.$$

(1)

This dynamical system is referred to as *the genotype dynamics* (for a detailed derivation, see Supplementary Information C in [3]). The foundation of dynamics (1) lies in the basic tenet of Darwinism: the relative frequency of a genotype increases if its production rate exceeds the average production rate of the population; in other words, when it has a positive relative advantage [42].

A key distinction between the replicator dynamics and the genotype dynamics is that the former describes asexual populations, where a parent of type $G_i$ produces only offspring of the same type, whereas dynamics (1) applies to sexual populations, where parents of type $G_i$ may produce offspring with genotypes different from $G_i$ (see Tables 1 and 4–6).

However, if the population is large enough such that all genotypes are present (i.e., $x_i > 0$ for every $i = 1, 2, \ldots, m$), then dynamics (1) can be rewritten in the following replicator-dynamics-like form:

$$\dot{x}_i = x_i \left( \frac{V_i(x)}{x_i} - \sum_j V_j(x) \right) = x_i \left( U_i(x) - \overline{U}(x) \right).$$

**Evolutionary stability of genotype distributions.** We now ask under what conditions a genotype distribution qualifies as an *evolutionarily stable genotype distribution* (ESGD) in an arbitrarily large sexual population. If every individual in the resident population has the same homozygous genotype, then, following the classical definition by Maynard Smith and Price [43], this genotype is said to be evolutionarily stable if no mutant genotype can invade the population through natural selection. Mathematically, this means that the (relative) frequency of the resident homozygotes increases from generation to generation, provided mutations are rare. The formal definition of ESGD for such homozygous populations was introduced in [3].

We now extend this concept to mixed genotype distributions. However, the original notion by Maynard Smith and Price [43] is not directly applicable when the genotype population exhibits polymorphism. Adopting a dynamic stability perspective, we define a polymorphic state as an ESGD if the system returns to this state after a small perturbation, such as the appearance of a rare mutant [44]. From a static, evolutionary viewpoint, a genotype distribution is considered to be an ESGD if the average production rate of the entire genotype population (as on the right-hand side of inequality (2) below) is strictly less than that of the genotype subpopulation in the ESGD state (on the left-hand side of inequality (2)). In other words, the ESGD subpopulation enjoys a relative advantage over the total population.

**Definition 1**. *A genotype distribution $x^* \in$ int $S_m$ is evolutionary stable if*

$$\sum_i x_i^* U_i(x) > \overline{U}(x),$$
(2)

*provided that $x$ is sufficiently close to (but not equal to) $x^*$.*

Clearly, inequality (2) can be rewritten as

$$\sum_i \frac{x_i^*}{x_i} V_i(x) > \sum_i V_i(x),$$
(3)

and it follows that $x^* \in \text{int} S_m$ is an ESGD if and only if

$$F(x^*, x) = \sum_i (x_i^* - x_i) \frac{V_i(x)}{x_i} > 0,$$
(4)

whenever $x$ is sufficiently close to (but not equal to) $x^*$. Note that $V_i(x)/x_i$ represents the average production rate of genotype $G_i$ and thus $V_i(x)/x_i$ plays a role analogous to that of fitness in asexual populations.

In asexual populations, coexisting types at an evolutionarily stable state exhibit equal fitness. A similar condition holds in our model. In section SI A of S1 File, we show that the following equilibrium condition (interpretable as a Nash equilibrium in game theoretical terms) holds.

**Equilibrium condition.** If $x^* \in \text{int} S_m$ is an ESGD and $x$ is an arbitrary state in $S_m$ then

$$\sum_{i=1}^m (x_i^* - x_i) \frac{V_i(x^*)}{x_i^*} \geq 0.$$
(5)

Furthermore, a key property of interior Nash equilibria in asexual matrix games also holds in this model. Namely, at an interior ESGD, the average production rates of all genotypes must be equal. Formally, if $x^* \in \text{int} S_m$ is an ESGD, then for every $i, j \in \{1, 2, \ldots, m\}$ we have

$$\frac{V_i(x^*)}{x_i^*} = \frac{V_j(x^*)}{x_j^*}.$$
(6)

This equality implies that an ESGD is a rest point of the genotype dynamics (1).

Following the reasoning in Chapter 7.2 of [41], we can establish the ensuing result (see Theorem SI.1 in section SI B of S1 File).

**Theorem 1**. *Assume that $x^* \in \text{int} S_m$ is an ESGD. Then $x^*$ is a locally asymptotically stable rest point of the genotype dynamics* (1).

We also need to define ESGD for a vertex of the simplex, particularly when all individuals share the same homozygous genotype. This definition was introduced in Supplementary Information B in [3], and we recall here. We say that $x^* = (1, 0, ..., 0)$ is an ESGD if

$$V_1(x) > x_1 \sum_j V_j(x)$$

for every $x$ sufficiently close to (but not equal to) $x^*$. (We note that equation (4) remains applicable if we agree that, in cases where $x_i^* = x_i = 0$, the expression $x_i^*/x_i$ is interpreted as 0 while $x_i/x_i$ is interpreted as 1.) It has been shown that if the vertex $x^* = (1, 0, ..., 0)$ is an ESGD, then it is a locally asymptotically stable equilibrium point of the genotype dynamics (1) (see Supplementary Information C in [3]).

## Application: Filial selection with a matrix game between full siblings

We consider a diploid species with internal fertilization. The following assumptions are made: mutations are sufficiently rare; the species practices monogamy, resulting in a relatedness of 1/2 between full siblings; and mating is *panmictic*, meaning that the probability of mating between genotypes $i$ and $j$ is $h_{ij}(x) = x_i x_j$. The population size $N$ is assumed to be very large, with $N/2$ mating pairs are formed randomly, ensuring random mating and preventing inbreeding.

As stated in the introduction, our goal is to generalize Haldane's concept of "*familial selection*" [1]. For this purpose, we adopt the simplifying assumption that each mating pair produces a fixed number $n$ of offspring, that is, $o_{ij}(x) = n$ for all $i, j$. Furthermore, an offspring's probability of survival depends solely on the behaviour of its siblings and is independent of the overall genotype distribution in the population. These simplifications ensure that the frequency-dependent interaction between mating pairs and the population-dependent juvenile survival rate cannot obscure the effect of the survival game between full siblings. As a result of differing survival probabilities among genotypes, the diploid parental population generally deviates from Hardy-Weinberg equilibrium.

For simplicity, we focus on a single autosomal locus with two alleles, denoted by [a] and [A], resulting in three possible genotypes, $G_1 = ([a],[a])$, $G_2 = ([a],[A])$, and $G_3 = ([A],[A])$ with corresponding frequencies $x_1$, $x_2$ and $x_3$. Thus the population state is represented by the vector $x = (x_1, x_2, x_3)$, which belongs to the simplex

$$S_3 := \{(u_1, u_2, u_3) : u_1 \geq 0, u_2 \geq 0, u_3 \geq 0, u_1 + u_2 + u_3 = 1\}.$$

*Genotype-phenotype mapping.* We assume that each genotype uniquely determines a phenotype. Let $s_1$, $s_2$ and $s_3$ denote the phenotypes associated with genotypes $G_1 = ([a],[a])$, $G_2 = ([a],[A])$ and $G_3 = ([A],[A])$, respectively. We consider both dominant-recessive and intermediate Mendelian inheritance systems.

We assume that all interactions relevant to the model occur between full siblings and are described by the same two-player one-shot matrix game, where the entries of the payoff matrix represent survival probabilities. Each sibling's survival rate is then the average payoff from interactions with its siblings. Accordingly, we consider a two-dimensional matrix game within each family, with a common payoff matrix $A \in R^{2 \times 2}$,

$$A = \begin{pmatrix} a_{11} & a_{12} \\ a_{21} & a_{22} \end{pmatrix}$$

where $a_{ij} \in (0, 1]$ for every $i, j \in \{1, 2\}$.

The phenotypes are pure or mixed strategies from the simplex

$$S_2 := \left\{ s = \left(s^{(1)}, s^{(2)}\right) \in R^2 : s^{(1)} \geq 0, s^{(2)} \geq 0, s^{(1)} + s^{(2)} = 1 \right\}.$$

Under recessive-dominant inheritance, each individual adopts a pure strategy, either $(1, 0)$ or $(0, 1)$. Under intermediate inheritance, the homozygotes follows a pure strategy, while each heterozygote employs the mixed strategy$(\frac{1}{2}, \frac{1}{2})$. Note that the survival rate of an offspring depends solely on its own phenotype and the phenotypes of its full siblings, which are determined by the parents' genotypes. Consequently, the family unit acts as a genetically well-defined interacting group, thereby introducing a "group selection ingredient" to our model.

**Arbitrary inheritance.** First, we give an overview of payoffs in the general case for two alleles at a single autosomal locus (see Table 1). Genotype $G_1 = ([a],[a])$ is associated with phenotype $s_1$, genotype $G_2 = ([a],[A])$ with phenotype $s_2$ and genotype $G_3 = ([A],[A])$ with phenotype $s_3$. That is, all heterozygotes exhibit the same phenotype, irrespective of the parental origin of each allele.

Above we defined a selection regime in Haldane's familial selection framework, where the interaction between full siblings is modelled by a two-player matrix game. In this setting, the expected number of surviving offspring of genotype $G_k$ from a $G_i \times G_j$ family, denoted by $n_{k(ij)}$, is summarized in Table 1.

Within each family, the game theoretical interaction is assumed to be well-mixed. This means that the genotypes of the offspring are independent of one another, and siblings interact in randomly formed pairs. Thus, in any interacting pair, the genotypes of the two individuals are independent. For instance, in a $G_2 \times G_2$ family, a zygote has genotype $G_1$, $G_2$ and $G_3$ with probabilities 1/4, 1/2 and 1/4, respectively. As a result, a juvenile of genotype $G_1$ interacts with phenotypes $s_1$, $s_2$ and $s_3$ with probabilities 1/4, 1/2 and 1/4. Consequently, the survival rate of a focal $G_1$ juvenile is given by

$$\frac{1}{4}s_1As_1 + \frac{1}{2}s_1As_2 + \frac{1}{4}s_1As_3.$$

Since the expected proportion of $G_1$ offspring in a $G_2 \times G_2$ family is 1/4, the average (expected) number of surviving $G_1$ juveniles in such families is

$$n_{1(22)} = \frac{n}{16}\left(s_1As_1 + 2s_1As_2 + s_1As_3\right).$$

(For further details, see section SI C of S1 File.)

Based on Table 1, the total number of individuals of genotypes $G_1$, $G_2$ and $G_3$ in the next generation can be calculated as follows:

$$V_1(x) = \frac{nN}{2}\left(x_1^2 s_1As_1 + \frac{x_1x_2}{2}\left(s_1As_1 + s_1As_2\right) + \frac{x_2^2}{16}\left(s_1As_1 + 2s_1As_2 + s_1As_3\right)\right),$$

$$V_2(x) = \frac{nN}{2}\left(\frac{x_1x_2}{2}\left(s_2As_1 + s_2As_2\right) + 2x_1x_3s_2As_2 + \frac{x_2^2}{8}\left(s_2As_1 + 2s_2As_2 + s_2As_3\right) + \frac{x_2x_3}{2}\left(s_2As_2 + s_2As_3\right)\right),$$

$$V_3(x) = \frac{nN}{2}\left(\frac{x_2^2}{16}\left(s_3As_1 + 2s_3As_2 + s_3As_3\right) + \frac{x_2x_3}{2}\left(s_3As_2 + s_3As_3\right) + x_3^2 s_3As_3\right).$$

Using these genotype-specific production rates, we obtain the explicit form of the genotype dynamics (1) under the selection regime considered.

We focus on the following questions in the general case when the phenotype of $G_1$ homozygotes is (1,0) while the phenotype of $G_3$ homozygotes is (0,1).

**Question 1**. What is a necessary condition for the stability of the homozygote sates?

(Here we present only the first-order condition. The detailed mathematical analysis is provided in section SI C of S1 File.) Let allele [a] be recessive and allele [A] dominant.

The state (1,0,0), in which every individual is a recessive homozygote $G_1 = ([a],[a])$, is an ESGD if

$$2a_{11} > a_{21} + a_{22}. \tag{8}$$

The state (0,0,1), in which every individual is a dominant homozygote $G_3 = ([A],[A])$, is an ESGD if

$$a_{22} > \frac{1}{5}\left(a_{11} + 3a_{12} + a_{21}\right). \tag{9}$$

**Question 2**. What is a necessary condition for the coexistence of all the genotypes?

Clearly, the homozygote states are the only possible rest points of the genotype dynamics on the boundary of the simplex $S_3$, and $S_3$ is positively invariant under the genotype dynamics (see Supplementary Information C in [3]). Therefore, by the two-dimensional Poincaré-Bendixson theorem, if both homozygote states are repellors, then (in biological terms) the system exhibits either stable or cyclic coexistence of the two phenotypes.

The state (1,0,0), where all individuals are recessive homozygotes $G_1$ is a repellor (see Theorem SI.5 in section SI C of S1 File), if

$$2a_{11} < a_{21} + a_{22}. \tag{10}$$

The state (0,0,1), where all individuals are dominant homozygotes $G_3$, is a repellor (see Theorem SI.6 in section SI C of S1 File), if

$$a_{22} < \frac{1}{5}\left(a_{11} + 3a_{12} + a_{21}\right). \tag{11}$$

We note that if inequalities (10) and (11) are both satisfied, then neither homozygote state is an ESGD. Based on our earlier results (see Supplementary Information C in [3]), this implies that both are repellor. Thus, the combination of these two inequalities provides a sufficient condition for the coexistence of all three genotypes (for additional details, see section SI D in S1 File). It is worth noting that conditions (8)-(11) apply to arbitrary payoff matrices, provided that allele [a] is recessive.

**Remark 3.** For the reader's convenience, we provide a helpful guide for analysing an arbitrary payoff matrix. Condition (2) of ESGD in Definition 1 is non-linear in the genotype distribution (see Supplementary Information C in [3]). When the linear term of the function $F(x^*, x)$ (see equation (4)) is non-zero, its sign determines the evolutionary stability of the corresponding homozygote: a positive sign implies local stability, while a negative one implies instability. If the linear term vanishes, the second-order terms must be examined, which requires more detailed analysis.

Although one might consider this latter case to be exceptional and therefore less important, this is not so. As already observed in [3], if allele [a] is dominant, the linear term vanishes near the homozygote vertex (1,0,0). This occurs because, in that neighbourhood the most common family type is $G_1 \times G_1$, followed by $G_1 \times G_2$, with all other families being rare. The genotype production rates in this neighbourhood therefore depend primarily on interactions within these two family types. However, due to the dominance of allele [a], all individuals in both families express phenotype $s_1$ (since $s_1 = s_2$), causing the first-order term to cancel. A similar argument applies when allele [A] is dominant: the linear term then vanishes near the homozygote vertex (0,0,1).

This phenomenon is reminiscent of the classical definition of ESS for matrix games, where the second-order condition becomes necessary to determine the evolutionary stability of a mixed Nash equilibrium (see condition (6.12) on p 63 in [41]).

Since second-order conditions are typically quadratic in the entries of the payoff matrix, it is useful to identify simple linear inequalities that imply (or at least suggest) that a vertex is a repellor (see Tables 2 and 3). This practical need was a key motivation behind our introduction of genotype dynamics: the use of differential equations allows a more effective analysis of the stability of the vertices.

The results presented in Tables 2 and 3 are derived in SI C of S1 File and build on our earlier work in [3], where we provided first- and second-order conditions (in the form of inequalities) the evolutionary stability of a genotype. If the inequality in the first-order condition is reversed, the corresponding vertex becomes a repellor. However, this logic does *not* extend to second-order conditions: reversing such an inequality provides only a necessary, but not sufficient, condition

**Table 2. Conditions when states (1,0,0) and (0,0,1), respectively, are ESGD-s.**

| [a] is | Genotype $G_1$ is ESGD if | Genotype $G_3$ is ESGD if |
|---|---|---|
| Recessive | either $2a_{11} > a_{21} + a_{22}$, <br> or $2a_{11} = a_{21} + a_{22}$, $a_{11} \geq 2a_{22}$ and $a_{11} + 4a_{12} > 3a_{22}$, <br> or $2a_{11} = a_{21} + a_{22}$, $a_{11} < 2a_{22}$ and $7a_{11} \leq 2a_{12}$. | $a_{22} > \frac{1}{5}(a_{11} + 3a_{12} + a_{21})$ |
| Dominant | $a_{11} > \frac{1}{5}(a_{12} + 3a_{21} + a_{22})$ | either $2a_{22} > a_{11} + a_{12}$, <br> or $2a_{22} = a_{11} + a_{12}$, $a_{22} \geq 2a_{11}$ and $a_{22} + 4a_{21} > 3a_{11}$, <br> or $2a_{22} = a_{11} + a_{12}$, $a_{22} < 2a_{11}$ and $7a_{22} \leq 2a_{21}$. |
| Intermediate | either $a_{11} > \frac{1}{5}(a_{12} + 3a_{21} + a_{22})$, <br> or $a_{11} = \frac{1}{5}(a_{12} + 3a_{21} + a_{22})$ and $2a_{12} \geq a_{21} + 3a_{22}$. | either $a_{22} > \frac{1}{5}(a_{11} + 3a_{12} + a_{21})$, <br> or $a_{22} = \frac{1}{5}(a_{11} + 3a_{12} + a_{21})$ and $2a_{21} \geq 3a_{11} + a_{12}$. |

(1,0,0) is the state in which the genotype of every individual is $G_1 = ([a],[a])$ with phenotype (1,0). (0,0,1) is the state in which the genotype of every individual is $G_3 = ([A],[A])$ with phenotype (0,1). [a] and [A] denote different alleles of the same gene. $(a_{ij})_{2\times 2}$ is the 2×2 payoff matrix describing the interactions between siblings.

**Table 3. Conditions when states (1,0,0) and (0,0,1), respectively, are repellors.**

| [a] is | Genotype $G_1$ is not ESGD | Genotype $G_3$ is not ESGD |
|---|---|---|
| Recessive | $G_1$ is a repellor if $2a_{11} < a_{21} + a_{22}$. <br> $G_1$ is suspected of being a repellor if $2a_{11} = a_{21} + a_{22}$ and $a_{11} + 4a_{12} < 3a_{22}$. | $G_3$ is suspected of being a repellor if $a_{22} < \frac{1}{5}(a_{11} + 3a_{12} + a_{21})$. |
| Dominant | $G_1$ is suspected of being a repellor if $a_{11} < \frac{1}{5}(a_{12} + 3a_{21} + a_{22})$. | $G_3$ is a repellor if $2a_{22} < a_{11} + a_{12}$. <br> $G_3$ is suspected of being a repellor if $2a_{22} = a_{11} + a_{12}$ and $a_{22} + 4a_{21} < 3a_{11}$. |
| Intermediate | $G_1$ is a repellor if $a_{11} < \frac{1}{5}(a_{12} + 3a_{21} + a_{22})$. <br> $G_1$ is suspected of being a repellor if $a_{11} = \frac{1}{5}(a_{12} + 3a_{21} + a_{22})$ and $3a_{12} + a_{21} \leq 2a_{22}$. | $G_3$ is a repellor if $a_{22} < \frac{1}{5}(a_{11} + 3a_{12} + a_{21})$. $G_3$ is suspected of being a repellor if $a_{22} = \frac{1}{5}(a_{11} + 3a_{12} + a_{21})$ and $a_{12} + 3a_{21} \leq 2a_{11}$. |

(1,0,0) is the state in which the genotype of every individual is $G_1 = ([a],[a])$ with phenotype (1,0). (0,0,1) is the state in which the genotype of every individual is $G_3 = ([A],[A])$ with phenotype (0,1). [a] and [A] denote different alleles of the same gene. $(a_{ij})_{2\times 2}$ is the 2×2 payoff matrix describing the interactions between siblings. The term "suspected" is used to indicate that the condition is necessary, but its sufficiency has not been proven.

for instability. Since the inequalities in Table 3 stem from second-order considerations, they merely *suggest* that the vertex may be a repellor, but further analysis is required to confirm this.

In what follows, we illustrate how the joint effect of the phenotypic payoff matrix and the genotype-phenotype mapping determines the solution of the genotype dynamics. It is important to note that our aim is not to provide a rigorous topological classification for the genotype dynamics, but rather to present numerical examples that offer insights into the underlying mechanisms.

To this end, we assume that the interaction between siblings is modelled by a prisoner's dilemma game, in which the two pure strategies are cooperation and defection. The corresponding payoff matrix is

|  | Cooperate | Defect |
|---|---|---|
| Cooperate | $a_{11}$ | $a_{12}$ |
| Defect | $a_{21}$ | $a_{22}$ |

where $a_{21} > a_{11} > a_{22} > a_{12}$.

From the viewpoint of group selection, we distinguish two subclasses of the prisoner's dilemma depending on how siblings can maximize the "welfare of the family" (i.e., the sum of their payoffs).

*Coordinated case*: If $2a_{11} > a_{21} + a_{12}$, then both siblings should cooperate to maximize their total survival rate. Note that although this condition is required in the iterated version of the game, our focus here is solely on the one-shot interaction.

*Anti-coordinated case:* If $2a_{11} < a_{21} + a_{12}$, then the highest collective payoff is achieved when one sibling cooperates and the other defects.

These terms are introduced in analogy with the coordination game (in which each player's individual payoff is maximized when both use the same pure strategy) and the anti-coordination game (in which individual payoffs are maximized when the players use different pure strategies), as the underlying ideas show a clear conceptual parallel (cf., [45]). However, the cases are not identical: in our setting, the joint payoff of the players is what matters, whereas in coordination and anti-coordination games, the focus is on maximizing individual payoffs.

A key question of the article is whether the evolutionary process indeed promotes cooperation in the coordinated case, or a division of roles (cooperator and defector) in the anti-coordinated case. As shown in Figs 10 and 11, which summarize the main subcases, the long-term outcomes do not necessarily align with the intuitive expectations suggested by these labels. For example, in the coordinated case, it is possible that only the defector homozygote state is stable. Unfortunately, we are unable to provide an intuitive explanation for this outcome, though it does emerge in analytical investigations.

We emphasize that in the inequalities above, the multiplier 2 refers the number of players in the interaction, and is not related to the coefficient of genetic relatedness between full siblings.

In the next subsection we analyse how the outcome of the familial selection model depends on whether the cooperator phenotype is recessive, dominant or intermediate.

**The cooperator phenotype is recessive.** If the cooperator phenotype is recessive then $s_1 = (1,0)$ and $s_2 = s_3 = (0,1)$. For the corresponding table, see Table 4.

In this setting, inequalities (8) and (9) ensure the evolutionary (and so asymptotic) stability of the homozygous states $(1,0,0)$ and $(0,0,1)$ while inequalities (10) and (11) provide sufficient conditions for the stable coexistence of all genotypes. Based on these conditions, four subcases can be identified in the coordinated case: both homozygous states are stable, only one is stable, or both of them are unstable. In contrast, in the anti-coordinated case, state $(1,0,0)$ can never be stable so only two subcases are possible: $(1,0,0)$ is unstable and $(0,0,1)$ is stable or both homozygous states are unstable. These subcases are illustrated by numerical examples in Figs 1 and 2.

Although coexistence does not necessarily imply the existence of an interior equilibrium, every numerical example in this article that satisfies the conditions for coexistence exhibits an (interior) ESGD. This demonstrates that Definition 1 is non-empty. To illustrate this, consider the example of coexistence from Fig 2. The payoff matrix is

**Table 4. Genotype survival table based on Table 1 if the cooperator strategy is recessive.**

| Genotypes of parents | Average number of couples assuming panmixia | Average number of surviving offspring, for genotypes $G_1$, $G_2$, $G_3$ | | |
|---|---|---|---|---|
| | | $G_1 = ([a],[a])$ recessive $s_1 = (1,0)$ | $G_2 = ([a],[A])$ dominant $s_2 = s_3 = (0,1)$ | $G_3 = ([A],[A])$, dominant $s_3 = (0,1)$ |
| $G_1 \times G_1$ | $\frac{N}{2}x_1^2$ | $n_{1(11)} = na_{11}$ | 0 | 0 |
| $G_1 \times G_2$ | $Nx_1x_2$ | $n_{1(12)} = \frac{n}{4}(a_{11} + a_{12})$ | $n_{2(12)} = \frac{n}{4}(a_{21} + a_{22})$ | 0 |
| $G_1 \times G_3$ | $Nx_1x_3$ | 0 | $n_{2(13)} = na_{22}$ | 0 |
| $G_2 \times G_2$ | $\frac{N}{2}x_2^2$ | $n_{1(22)} = \frac{n}{16}(a_{11} + 3a_{12})$ | $n_{2(22)} = \frac{n}{8}(a_{21} + 3a_{22})$ | $n_{3(22)} = \frac{n}{16}(a_{21} + 3a_{22})$ |
| $G_2 \times G_3$ | $Nx_2x_3$ | 0 | $n_{2(23)} = \frac{n}{2}a_{22}$ | $n_{3(23)} = \frac{n}{2}a_{22}$ |
| $G_3 \times G_3$ | $\frac{N}{2}x_3^2$ | 0 | 0 | $n_{3(33)} = na_{22}$ |

The phenotype corresponding to the genotype $G_1$ is the cooperator strategy, represented by $s_1 = (1,0)$. The phenotypes of genotypes $G_2$ and $G_3$ correspond to the defector strategy, represented by $s_3 = (0,1)$. Allele [a] is recessive, while [A] is dominant. $p_{k(ij)}$ denotes the probability that an offspring has genotype $G_k$, given that the parental genotypes are $G_i$ and $G_j$, respectively. $n_{k(ij)}$ denotes the number of surviving offspring with genotype $G_k$ provided the parental genotypes are $G_i$ and $G_j$, respectively. The matrix $(a_{ij})_{2\times2}$ matrix describes the payoff from interactions between siblings. $N$ is the size of the parental population. $n$ denotes the fixed number of newborn offspring per family.

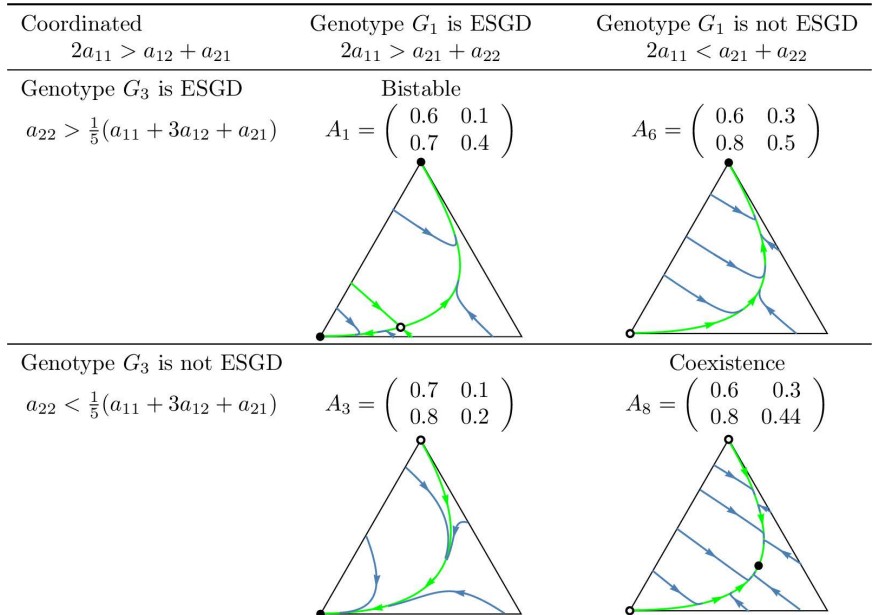

**Fig 1. Examples of prisoner's dilemma games in which cooperation is *recessive* and the total payoff is maximized by the *cooperator-cooperator* strategy pair (coordinated case).** Black dots indicate asymptotically stable rest points, while empty dots represent unstable equilibria. The bottom-left vertex of the triangle corresponds to the state $(1,0,0)$, in which all individuals are cooperator homozygotes with genotype $([a],[a])$. The bottom-right vertex represents the state $(0,1,0)$, where all individuals are heterozygotes with genotype $([a],[A])$. The top vertex corresponds to the state $(0,0,1)$, where all individuals are defector homozygotes with genotype $([A],[A])$.

$$A_{11} = \begin{pmatrix} 0.4 & 0.1 \\ 0.8 & 0.2 \end{pmatrix}.$$

The corresponding interior Nash equilibrium, which satisfies (6), is $(x_1^*, x_2^*, x_3^*) = (0.251, 0.56, 0.189)$. This equilibrium is evolutionarily stable, as shown in the left panel of Fig 3, where the function $F(x^*, x)$ is plotted in $x$ around $x^*$, clearly indicating

| Anti-coordinated $2a_{11} < a_{12} + a_{21}$ | Genotype $G_1$ is ESGD $2a_{11} > a_{21} + a_{22}$ | Genotype $G_1$ is not ESGD $2a_{11} < a_{21} + a_{22}$ |
|---|---|---|
| Genotype $G_3$ is ESGD $a_{22} > \frac{1}{5}(a_{11} + 3a_{12} + a_{21})$ | Never in alternating PD | $A_9 = \begin{pmatrix} 0.4 & 0.1 \\ 0.75 & 0.3 \end{pmatrix}$  |
| Genotype $G_3$ is not ESGD $a_{22} < \frac{1}{5}(a_{11} + 3a_{12} + a_{21})$ | Never in alternating PD | Coexistence $A_{11} = \begin{pmatrix} 0.4 & 0.1 \\ 0.8 & 0.2 \end{pmatrix}$  |

**Fig 2. Examples of prisoner's dilemma games in which cooperation is *recessive* and the total payoff is maximized by the *cooperator-defector* strategy pair (anti-coordinated case).** See the caption at Fig 1.

that $F(x^*, x)$ attains a minimum in variable $x$ at $x = x^*$. The right panel of Fig 1 shows the trajectories of the dynamics (1), demonstrating that the interior equilibrium point $x^*$ is asymptotically stable, whereas the two boundary equilibria $(1, 0, 0)$ and $(0, 0, 1)$ are unstable.

**The cooperator phenotype is dominant.** If the cooperator phenotype is dominant then $s_1 = s_2 = (1, 0)$ and $s_3 = (0, 1)$. For the corresponding table, see Table 5.

Observe that Tables 4 and 5 illustrate different selection scenarios. For instance, in $G_2 \times G_2$ families, the survival rates vary. In particular, for each $k$, the value of $n_{k(22)}$ differs. According to the genotype-phenotype mapping, if the cooperator phenotype is dominant, the number of cooperators will be higher in families with the same genotype distribution.

If allele [a] responsible for cooperation is dominant, then the conditions of the stability of the homozygous states are as follows (see section SI C in S1 File).

State $(1, 0, 0)$, in which all individuals are homozygotes $G_1 = ([a],[a])$, is an ESGD if

$$a_{11} > \frac{1}{5}\left(a_{12} + 3a_{21} + a_{22}\right) \tag{12}$$

State $(0, 0, 1)$, in which all individuals are homozygotes $G_3 = ([A],[A])$, is an ESGD if

$$2a_{22} > a_{11} + a_{12} \tag{13}$$

Similarly to the recessive case, reversing the inequality (12) ensures that state $(1, 0, 0)$ is a repellor and not an ESGD. Likewise, reversing inequality (13) implies that state $(0, 0, 1)$ is a repellor and not an ESGD. It is important to note that conditions (12) and (13) apply to arbitrary payoff matrices, provided that allele [a] is dominant.

When the interaction among siblings is described by the prisoner's dilemma, we can again distinguish four subcases in the coordinated case and two subcases in the anti-coordinated case, based on the stability properties of states $(1, 0, 0)$ and $(0, 0, 1)$. These subcases are illustrated with particular examples in Figs 4 and 5.

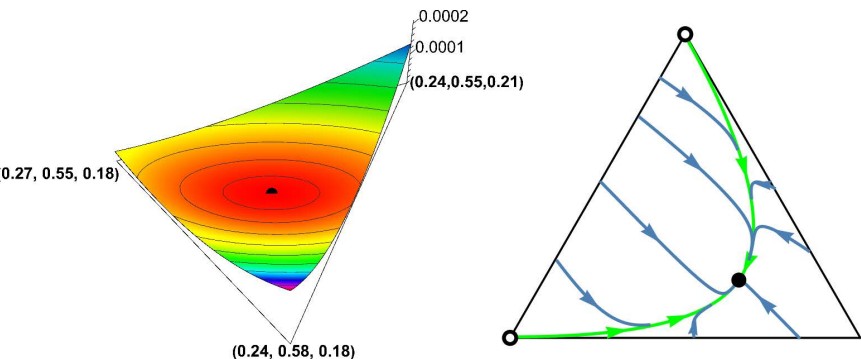

**Fig 3. Definition 1 is not empty.** *Left panel*: *The function* $\mathbf{F}(x^*, x)$ *around* $\mathbf{x} = \mathbf{x}^* = (0.251, 0.56, 0.189)$. The black dot marks the graph point $(x^*, F(x^*, x^*))$ corresponding to $\mathbf{x}^*$. It is clearly visible that $\mathbf{F}(x^*, x)$ in variable $\mathbf{x}$ attains a minimum at this point. *Right panel: Phase portrait of dynamics (1)*. The system is governed by the phenotypic payoff matrix $\mathbf{A}_{11} = \begin{pmatrix} 0.4 & 0.1 \\ 0.8 & 0.2 \end{pmatrix}$ from Fig 2. The interior rest point $\mathbf{x}^* = (0.251, 0.56, 0.189)$ is globally asymptotically stable; the vertices $(\mathbf{1,0,0})$ and $(\mathbf{0,0,1})$ corresponding to monomorphic population of homozygotes are unstable rest points. Black dots and circles indicate asymptotically stable and unstable rest points, respectively. Note that the simplex $\mathbf{S}_3$ is only positively invariant under the genotype dynamics (1), in general. For instance, while the homozygote vertices (1,0,0) and (0,0,1) are trivial rest points of the genotype dynamics (1), the heterozygote vertex (0,1,0) is not. It becomes a rest point of the genotype dynamics (1) if and only if both homozygotes are lethal.

**Table 5. Genotype survival table based on Table 1 if the cooperator strategy is dominant.**

| Genotypes of parents | Average number of couples assuming panmixia | Average number of surviving offspring, for genotypes $G_1$, $G_2$, $G_3$ | | |
|---|---|---|---|---|
| | | $G_1 = ([a],[a])$ dominant $s_1 = (1,0)$ | $G_2 = ([a],[A])$ dominant $s_2 = s_1 = (1,0)$ | $G_3 = ([A],[A])$, recessive $s_3 = (0,1)$ |
| $G_1 \times G_1$ | $\frac{N}{2}x_1^2$ | $n_{1(11)} = na_{11}$ | 0 | 0 |
| $G_1 \times G_2$ | $Nx_1x_2$ | $n_{1(12)} = \frac{n}{2}a_{11}$ | $n_{2(12)} = \frac{n}{2}a_{11}$ | 0 |
| $G_1 \times G_3$ | $Nx_1x_3$ | 0 | $n_{2(13)} = na_{11}$ | 0 |
| $G_2 \times G_2$ | $\frac{N}{2}x_2^2$ | $n_{1(22)} = \frac{n}{16}(3a_{11} + a_{12})$ | $n_{2(22)} = \frac{n}{8}(3a_{11} + a_{12})$ | $n_{3(22)} = \frac{n}{16}(3a_{21} + a_{22})$ |
| $G_2 \times G_3$ | $Nx_2x_3$ | 0 | $n_{2(23)} = \frac{n}{4}(a_{11} + a_{12})$ | $n_{3(23)} = \frac{n}{4}(a_{21} + a_{22})$ |
| $G_3 \times G_3$ | $\frac{N}{2}x_3^2$ | 0 | 0 | $n_{3(33)} = na_{22}$ |

The phenotypes of genotypes $G_1$ and $G_2$ correspond to the cooperator strategy, represented by $s_1 = (1,0)$. the phenotype corresponding to genotype $G_3$ is the defector strategy, represented by $s_3 = (0,1)$. Allele [a] is dominant while [A] is recessive. $p_{k(ij)}$ denotes the probability that an offspring has genotype $G_k$, given that the parental genotypes are $G_i$ and $G_j$, respectively. $n_{k(ij)}$ denotes the number of surviving offspring with genotype $G_k$ provided the parental genotypes are $G_i$ and $G_j$, respectively. The matrix $(a_{ij})_{2\times2}$ describes the payoffs from interactions between siblings. $N$ is the size of the parental population. $n$ denotes the fixed number of newborn offspring per family.

**Intermediate inheritance.** Finally, we turn to the case of intermediate inheritance. In this scenario, the phenotype of the homozygotes $G_1 = ([a],[a])$ is the pure strategy $(1,0)$, the phenotype of the homozygotes $G_3 = ([A],[A])$ is the pure strategy $(0,1)$, while the phenotype of the heterozygotes $G_2 = ([a],[A])$ is the mixed strategy $\left(\frac{1}{2}, \frac{1}{2}\right)$ (see Table 6).

As in the recessive and the dominant cases, we provide mathematical conditions for the stability of the homozygote states $(1,0,0)$ and $(0,0,1)$. In the main text, we only present the first-order condition. The detailed mathematical analysis is available in section SI C of S1 File.

State $(1,0,0)$, in which all individuals are homozygotes $G_1 = ([a],[a])$, is an ESGD if

$$a_{11} > \frac{1}{5}\left(a_{12} + 3a_{21} + a_{22}\right).$$

(14)

| Coordinated $2a_{11} > a_{12} + a_{21}$ | Genotype $G_1$ is ESGD $a_{11} > \frac{1}{5}(a_{12} + 3a_{21} + a_{22})$ | Genotype $G_1$ is not ESGD $a_{11} < \frac{1}{5}(a_{12} + 3a_{21} + a_{22})$ |
|---|---|---|
| Genotype $G_3$ is ESGD $2a_{22} > a_{11} + a_{12}$ | Bistable $A_1 = \begin{pmatrix} 0.6 & 0.1 \\ 0.7 & 0.4 \end{pmatrix}$ | $A_7 = \begin{pmatrix} 0.3 & 0.1 \\ 0.49 & 0.21 \end{pmatrix}$ |
| Genotype $G_3$ is not ESGD $2a_{22} < a_{11} + a_{12}$ | $A_3 = \begin{pmatrix} 0.7 & 0.1 \\ 0.8 & 0.2 \end{pmatrix}$ | Coexistence $A_4 = \begin{pmatrix} 0.5 & 0.1 \\ 0.8 & 0.15 \end{pmatrix}$ |

**Fig 4. Examples of prisoner's dilemma games in which cooperation is *dominant* and the total payoff is maximized by the *cooperator-cooperator* strategy pair (coordinated case).** Black dots indicate asymptotically stable rest points, while empty dots represent unstable equilibria. The bottom-left vertex of the triangle corresponds to the state $(1,0,0)$, in which all individuals are cooperator homozygotes with genotype ([a],[a]). The bottom-right vertex represents the state $(0,1,0)$, where all individuals are heterozygotes with genotype ([a],[A]). The top vertex corresponds to the state $(0,0,1)$, where all individuals are defector homozygotes with genotype ([A],[A]).

State $(0, 0, 1)$, in which all individuals are homozygotes $G_3 = ([A],[A])$, is an ESGD if

$$a_{22} > \frac{1}{5}\left(a_{11} + 3a_{12} + a_{21}\right).$$

(15)

Reversing inequality (14) implies that state $(1, 0, 0)$ is a repellor and not an ESGD, and reversing inequality (15) implies that state $(0, 0, 1)$ is a repellor and not an ESGD. These conclusions hold for arbitrary matrices under intermediate inheritance.

When applying inequalities (14)-(15) to the context of the prisoner's dilemma, we again obtain four possible subcases in the coordinated case and two subcases in the anti-coordinated case. These scenarios are illustrated with concrete numerical examples in Figs 6 and 7.

**Donation game in diploid Mendelian population and classical Hamilton's rule**

A special case of the prisoner's dilemma is the so-called "donation game" [17], defined by the payoff matrix

$$\begin{pmatrix} b - c & -c \\ b & 0 \end{pmatrix},$$

where $b$ and $c$ denote the benefit and the cost of an interaction, respectively, with $b > c$. The donation game serves as a link between the prisoner's dilemma and Hamilton's rule [5,46,47]. Since in a monogamous diploid family the relatedness between full siblings is $r = 1/2$, the classical Hamilton's rule predicts cooperation between siblings when $b > 2c$.

| Anti-coordinated $2a_{11} < a_{12} + a_{21}$ | Genotype $G_1$ is ESGD $a_{11} > \frac{1}{5}(a_{12} + 3a_{21} + a_{22})$ | Genotype $G_1$ is not ESGD $a_{11} < \frac{1}{5}(a_{12} + 3a_{21} + a_{22})$ |
|---|---|---|
| Genotype $G_3$ is ESGD $2a_{22} > a_{11} + a_{12}$ | Never in alternating PD | $A_{10} = \begin{pmatrix} 0.4 & 0.1 \\ 0.9 & 0.3 \end{pmatrix}$ |
| Genotype $G_3$ is not ESGD $2a_{22} < a_{11} + a_{12}$ | Never in alternating PD | Coexistence $A_{11} = \begin{pmatrix} 0.4 & 0.1 \\ 0.8 & 0.2 \end{pmatrix}$ |

**Fig 5. Examples of prisoner's dilemma games in which cooperation is *dominant* and the total payoff is maximized by the *cooperator-defector* strategy pair (anti-coordinated case).** See the caption at Fig 4.

**Table 6. Genotype survival table based on Table 1 in case of intermediate inheritance.**

| Genotypes of parents | Average number of couples $\frac{N}{2}h_{ij}(x)$ | Average number of surviving offspring, for genotypes $G_1$, $G_2$, $G_3$ | | |
|---|---|---|---|---|
| | | $G_1 = ([a],[a])$ $s_1 = (1,0)$ | $G_2 = ([a],[A])$ $s_2 = \left(\frac{1}{2}, \frac{1}{2}\right)$ | $G_3 = ([A],[A])$ $s_3 = (0,1)$ |
| $G_1 \times G_1$ | $\frac{N}{2}x_1^2$ | $n_{1(11)} = na_{11}$ | $0$ | $0$ |
| $G_1 \times G_2$ | $Nx_1 x_2$ | $n_{1(12)} = \frac{n}{8}(3a_{11} + a_{12})$ | $n_{2(12)} = \frac{n}{16}(3a_{11} + a_{12} + 3a_{21} + a_{22})$ | $0$ |
| $G_1 \times G_3$ | $Nx_1 x_3$ | $0$ | $n_{2(13)} = \frac{n}{4}(a_{11} + a_{12} + a_{21} + a_{22})$ | $0$ |
| $G_2 \times G_2$ | $\frac{N}{2}x_2^2$ | $n_{1(22)} = \frac{n}{8}(a_{11} + a_{12})$ | $n_{2(22)} = \frac{n}{8}(a_{11} + a_{12} + a_{21} + a_{22})$ | $n_{3(22)} = \frac{n}{8}(a_{21} + a_{22})$ |
| $G_2 \times G_3$ | $Nx_2 x_3$ | $0$ | $n_{2(23)} = \frac{n}{16}(a_{11} + 3a_{12} + a_{21} + 3a_{22})$ | $n_{3(23)} = \frac{n}{8}(a_{21} + 3a_{22})$ |
| $G_3 \times G_3$ | $\frac{N}{2}x_3^2$ | $0$ | $0$ | $n_{3(33)} = na_{22}$ |

The phenotype corresponding to genotype $G_1$ is the cooperator strategy, represented by $s_1 = (1,0)$; the phenotype corresponding to genotype $G_3$ is the defector strategy, represented by $s_3 = (0,1)$; while the phenotype corresponding to the heterozygote genotype $G_2$ is $(1/2,\ 1/2)$, denoted by $s_2$. $p_{k(ij)}$ denotes the probability that an offspring has genotype $G_k$, given that the parental genotypes are $G_i$ and $G_j$, respectively. $n_{k(ij)}$ denotes the number of surviving offspring with genotype $G_k$ provided the parental genotypes are $G_i$ and $G_j$, respectively. The matrix $(a_{ij})_{2\times 2}$ describes the payoffs from interactions between siblings. $N$ is the size of the parental population. $n$ denotes the fixed number of newborn offspring per family.

Although the donation game is formally not a survival game (the entries of the payoff matrix are not survival probabilities), starvation can be one of the primary causes of mortality. In times of famine, when survival depends on reaching a minimum level of food intake during a critical period, siblings may enhance each other's survival through food sharing. Therefore, an additive model (where the outcomes of individual interactions are summed) is more suitable for modelling food sharing than a multiplicative model (where outcomes are multiplied).

| Coordinated $2a_{11} > a_{12} + a_{21}$ | Genotype $G_1$ is ESGD $a_{11} > \frac{1}{5}(a_{12} + 3a_{21} + a_{22})$ | Genotype $G_1$ is not ESGD $a_{11} < \frac{1}{5}(a_{12} + 3a_{21} + a_{22})$ |
|---|---|---|
| Genotype $G_3$ is ESGD $a_{22} > \frac{1}{5}(a_{11} + 3a_{12} + a_{21})$ | Bistable $A_1 = \begin{pmatrix} 0.6 & 0.1 \\ 0.7 & 0.4 \end{pmatrix}$ | $A_6 = \begin{pmatrix} 0.6 & 0.3 \\ 0.8 & 0.5 \end{pmatrix}$ |
| Genotype $G_3$ is not ESGD $a_{22} < \frac{1}{5}(a_{11} + 3a_{12} + a_{21})$ | $A_3 = \begin{pmatrix} 0.7 & 0.1 \\ 0.8 & 0.2 \end{pmatrix}$ | Coexistence $A_4 = \begin{pmatrix} 0.5 & 0.1 \\ 0.8 & 0.15 \end{pmatrix}$ |

**Fig 6. Examples of prisoner's dilemma games with *intermediate inheritance* and in which the total payoff is maximized by the *cooperator-cooperator* strategy pair (coordinated case).** Black dots indicate asymptotically stable rest points while empty dots represent unstable equilibrium points. The bottom-left vertex of the triangle corresponds to the state (1,0,0), in which all individuals are ([a],[a]) cooperator homozygotes. The bottom-right vertex represents the state (0,1,0), where all individuals are heterozygotes with genotype ([a],[A]). The top vertex corresponds to the state (0,0,1), where all individuals are defector homozygotes with genotype ([A],[A]).

| Anti-coordinated $2a_{11} < a_{12} + a_{21}$ | Genotype $G_1$ is ESGD $a_{11} > \frac{1}{5}(a_{12} + 3a_{21} + a_{22})$ | Genotype $G_1$ is not ESGD $a_{11} < \frac{1}{5}(a_{12} + 3a_{21} + a_{22})$ |
|---|---|---|
| Genotype $G_3$ is ESGD $a_{22} > \frac{1}{5}(a_{11} + 3a_{12} + a_{21})$ | Never in alternating PD | $A_9 = \begin{pmatrix} 0.4 & 0.1 \\ 0.75 & 0.3 \end{pmatrix}$ |
| Genotype $G_3$ is not ESGD $a_{22} < \frac{1}{5}(a_{11} + 3a_{12} + a_{21})$ | Never in alternating PD | Coexistence $A_{10} = \begin{pmatrix} 0.4 & 0.1 \\ 0.9 & 0.3 \end{pmatrix}$ |

**Fig 7. Examples of prisoner's dilemma games with *intermediate inheritance* and in which the total payoff is maximized by the *cooperator-defector* strategy pair (anti-coordinated case).** See the caption at Fig 6.

It is worth noting that only non-negative payoff matrices make sense in the model. However, it is not a real limitation because adding the same constant to all entries of the matrix does not affect the relative fitness and thus leaves game-theoretical predictions unchanged. Therefore, instead of using the above payoff matrix, we can equivalently use the shifted matrix:

$$H = \begin{pmatrix} \pi + b - c & \pi - c \\ \pi + b & \pi \end{pmatrix},$$

where $\pi > c$ can be interpreted as a baseline, interaction-independent payoff.

This raises the question: under what condition is the cooperator homozygotes $G_1$ locally stable in the game between diploid siblings? Since $b > c$, the payoff matrix $H$ satisfies $a_{21} > a_{11} > a_{22} > a_{12}$ and $2a_{11} > a_{21} + a_{12}$, indicating that the donation game corresponds to a special case of the prisoner's dilemma, one in which the cooperator-cooperator pair maximizes total payoff (i.e., the coordinated case).

It is easy to see that inequalities (8), (12) and (14) correspond to the classical Hamilton's rule ($b > 2c$), inequalities (9), (13), and (15) are equivalent to $b < 2c$. Therefore, within the inheritance systems considered, when survival is determined by the donation game, there is neither bistability nor coexistence. Only two outcomes are possible: if $b > 2c$ then the cooperator homozygotes $G_1$ become fixed through selection; if $b < 2c$ then the defector homozygotes $G_3$ do.

### Haldane monotonicity

Price [48] was interested in how the relative frequency of an allele changes between the parental and the offspring population. Hamilton's rule [5,6] and Haldane's arithmetic [4] describe conditions under which the frequency of the altruistic allele strictly increases in well-defined selection scenario, thereby ensuring the evolutionary stability of the population of altruistic homozygotes. (Haldane's arithmetic claims [4]: "*The relative frequency of the altruistic gene increases, if by 'self-sacrifice', it has 'rescued' on average more than one copy of itself in its lineage.*" We note that Haldane's arithmetic is equivalent to evolutionary stability—in ESGD sense—of recessive homozygotes in the one-person altruistic game. Moreover, this equivalence holds even when the interaction between siblings is governed by a non-linear game [4].) These rules are applicable when fitness can be expressed in terms of explicit cost and benefit parameters. However, in the case of prisoner's dilemma games, at least three parameters are typically needed, making the application of both Hamilton's rule and Haldane's arithmetic less straightforward.

Instead, we adopt a different approach to investigate when the frequency of a given allele increases along every trajectory of the genotype dynamics. Mathematically, this corresponds to the allele frequency functioning as a Lyapunov function. For simplicity, we focus on the case in which three genotypes are possible. The frequency of the allele [a] in homozygote $G_1$ is given by

$$g(x) = \frac{2x_1 + x_2}{2} = x_1 + \frac{x_2}{2}.$$

The derivative of function $g$ with respect to the genotype dynamics (1) is

$$DV(x) = \dot{x}_1 + \frac{\dot{x}_2}{2}.$$

Since the genotype dynamics satisfies $\sum \dot{x}_i = 0$, it follows that

$$DV(x) = \dot{x}_1 + \frac{\dot{x}_2}{2} - \frac{\sum \dot{x}_i}{2} = \frac{\dot{x}_1}{2} - \frac{\dot{x}_3}{2}.$$

Thus, the frequency of the [a] allele strictly increases if

$$\dot{x}_1 - \dot{x}_3 > 0.$$

The biological interpretation of this inequality is the following: we say that the genotype dynamics (1) *Haldane monotonic* if the relative advantage of a $G_1$ homozygote over the entire population is greater than that of a $G_3$ homozygote. The key difference between Hamilton's rule, Haldane's arithmetic, and Haldane monotonicity lies in their respective focuses. Hamilton's rule emphasizes the interaction between two siblings, Haldane's arithmetic considers how many parental genes are saved through the "self-sacrifice" of a cooperating sibling, whereas Haldane monotonicity centres on the relative advantage of a particular homozygote taking into account production rates within the entire population.

**Remark 4**. Mathematically, Haldane monotonicity provides only a sufficient condition for the global stability of a homozygous state; global stability can still arise even when Haldane monotonicity is not satisfied. For example, it is possible for a trajectory to initially move away from the homozygous vertex but later reverse direction and onverge to it (cf. Fig 9).

We first illustrate, through a numerical example, the existence of a matrix game in which the genotype dynamics is Haldane monotone, see Fig 8.

Second, one may ask whether the genotype dynamics is necessarily Haldane monotone when a donation game governs interaction between siblings? The answer is negative, see Fig 9. The intuitive explanation is as follows. When the whole population consists entirely of heterozygotes, Mendelian inheritance dictates that diploid embryos will follow Hardy-Weinberg proportions, i.e., 1/4, 1/2 and 1/4 are the respective relative frequencies of the recessive homozygotes, heterozygotes and dominant homozygotes. If phenotypic selection does not substantially alter these Hardy-Weinberg proportions, then the frequency of the cooperative allele cannot increase. The key insight is that changes in genotype frequencies are determined jointly by the genetic system and phenotypic selection. We note that the heterozygote vertex is a rest point of the genotype dynamics only in the extreme case where both homozygotes are lethal in all family types.

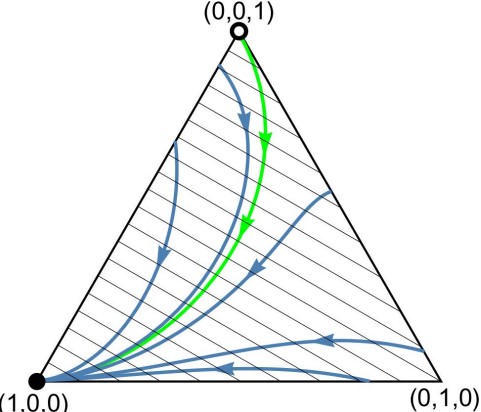

**Fig 8. Example for Haldane monotonicity.** Here we consider the payoff matrix $\begin{pmatrix} 0.85 & 0.35 \\ 0.6 & 0.1 \end{pmatrix}$ and assume that allele [a] is recessive to [A]. Notice that the orbits, as they move toward the vertex $(1,0,0)$ (representing the state where all individuals are recessive homozygotes ([a],[a])), intersect the level lines (displayed as parallel lines in the figure) of the function $x_1 + \frac{x_2}{2}$, which describes the frequency of the recessive allele, exactly once. The vertex $(0,1,0)$ corresponds to the state in which all individuals are heterozygotes ([a],[A]), while $(0,0,1)$ corresponds to the state in which all individuals are homozygotes ([A],[A]). The black dot indicates an asymptotically stable rest point, while the empty dot marks an unstable equilibrium.

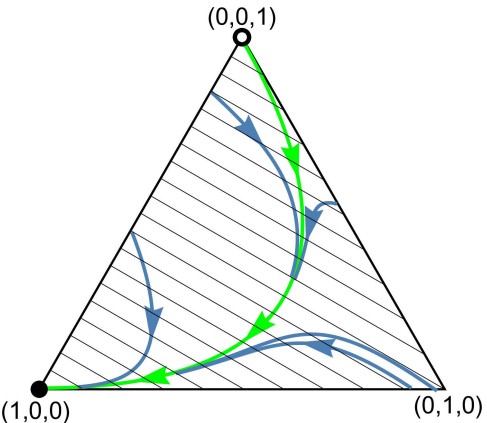

**Fig 9. Global stability of a homozygous state without Haldane monotonicity.** Consider a donation game with parameters $\pi = 0.2$, $b = 0.6$ and $c = 0.1$, leading to the payoff matrix $A_3 = \begin{pmatrix} 0.7 & 0.1 \\ 0.8 & 0.2 \end{pmatrix}$ and assume that allele [a] is recessive to allele [A]. The parallel lines are the level lines of the function $x_1 + \frac{x_2}{2}$, which describes the frequency of the recessive allele. Observe that, along the orbits starting near the heterozygote vertex $(0,1,0)$ (which is not an equilibrium), this function first decreases and then increases. This demonstrates that the dynamics is not globally Haldane monotone. However, if we exclude an appropriate neighbourhood of the pure heterozygote vertex, the system exhibits Haldane monotonicity. In particular, near the vertex $(1,0,0)$, where all individuals are of genotype ([a],[a]), the system is locally Haldane monotone. The vertex $(0,0,1)$ corresponds to the state in which all individuals are homozygotes with genotype ([A],[A]). The black dot indicates an asymptotically stable rest point, while the empty dot represents an unstable equilibrium.

**Table 7. Summary of the conditions on a payoff matrix $(a_{ij})_{2\times 2}$ for the homozygous states to be ESGD, and so asymptotically stable.**

|  | Genotype $G_1$= ([a],[a]) is ESGD | Genotype $G_3$=([A],[A]) is ESGD |
|---|---|---|
| [a] is recessive | $2a_{11} > a_{21} + a_{22}$ | $a_{22} > \frac{1}{5}\left(a_{11} + 3a_{12} + a_{21}\right)$ |
| [a] is dominant | $a_{11} > \frac{1}{5}\left(a_{12} + 3a_{21} + a_{22}\right)$ | $2a_{22} > a_{11} + a_{12}$ |
| Intermediate inheritance | $a_{11} > \frac{1}{5}\left(a_{12} + 3a_{21} + a_{22}\right)$ | $a_{22} > \frac{1}{5}\left(a_{11} + 3a_{12} + a_{21}\right)$ |

[a] and [A] denote different alleles of the same gene. (1,0,0) is the state in which all individuals have genotype $G_1 = ([a],[a])$ with phenotype (1,0). (0,0,1) is the state in which all individuals have genotype $G_3 = ([A],[A])$ with phenotype (0,1). The matrix $(a_{ij})_{2\times 2}$ represents a 2×2 payoff matrix describing the interactions between siblings.

## Discussion

### In diploid populations, the payoff matrix and the genotype-phenotype mapping together determine the evolutionary outcome

We first examine the conditions under which the homozygous states are evolutionarily stable, depending on the inheritance system and the (arbitrary) survival payoff matrix (i.e., where each matrix element is between 0 and 1). Applying the results in Supplementary Information B in [3] for matrix games, we obtain the following Table 7 (see SI C of S1 file).

Several noteworthy observations follow:

- The condition for the evolutionary stability of the dominant $G_3$ is the same as that for intermediate $G_3$ homozygote. Hence in the case of dominant-recessive and intermediate inheritance, the evolutionary stability (in ESGD sense), instability, or degeneracy of the state associated with $G_3$ coincide.

- Similarly, the condition for the dominant $G_1$ homozygote is identical to that for intermediate $G_1$ homozygote. Consequently, in both dominant and intermediate inheritance systems, the evolutionary stability, instability, or degeneracy of the state associated with $G_1$ coincide.

- It follows that the recessive and dominant cases (with respect to the allele in $G_1$) unambiguously determine the evolutionary outcome in the intermediate case. In particular, the stability of the state associated with $G_1$ in the intermediate case matches that in the dominant case, while the stability of the state associated with $G_3$ is the same as in the recessive case.

These coincidences are somewhat surprising, given the selection regimes defined in Tables 4–6 are quite different.

Secondly, based on these observations, we classify prisoner's dilemma payoff matrices according to the evolutionary stability of the two homozygous states under different inheritance systems. In the coordinated case, where the cooperator-cooperator strategy pair maximizes the total payoff of the siblings, we find eight distinct evolutionary scenarios. In the anti-coordinated case, where the cooperator-defector strategy pair maximizes the sum of the siblings' payoffs, there are only three. These cases are summarized in Figs 10 and 11, each accompanied by a concrete numerical example.

The numerical examples shown in Figs 10 and 11 highlight several important points:

- Both the cooperator and defector homozygote states can be the unique stable state by natural selection, see, e.g., $A_3$ and $A_6$, respectively.

- Bistable scenarios can arise (e.g., $A_1$) and, in such cases, there is exactly one unstable interior rest point.

- Cooperator and defector genotypes can coexist (e.g., $A_8$), with a globally stable interior rest point in such cases. The genotype-phenotype map can change the coordinates of this stable rest point of the genotype dynamics.

- Although the phenotypic payoff matrix alone is not sufficient to predict the asymptotic stability of cooperation, it plays a decisive role in shaping the evolutionary outcome. For instance, in the anti-coordinated case, the population of cooperator homozygotes cannot be stable, yet coexistence between cooperators and defectors is still possible (see $A_{11}$).

- Despite the 1/2 genetic relatedness between siblings, and even when cooperation maximizes the total survival probability of siblings in the coordinated case, the population of defector homozygotes can be the unique asymptotically stable rest point of the genotype dynamics (e.g., $A_6$).

- The phenotypic payoff matrix and the genotype-phenotype mapping together determine the outcome of the natural selection. When the phenotypic payoff matrix is fixed, the genotype-phenotype mapping cannot only influence the coordinates of the stable rest point of the genotype dynamics, but can also determine whether both homozygotes are present or one of them is absent in the stable equilibrium (e.g., $A_{10}$).

- However, there is no case in which one inheritance system exhibits bistability while another leads to coexistence. At most, the stability of a single homozygous vertex can change. This suggests that although the genotype-phenotype mapping can influence the outcome of natural selection, the payoff matrix still imposes structural constraints on evolutionary possibilities.

## Our results from the viewpoint of group selection

In our classification of prisoner's dilemma games, the principle of "family welfare," which means that siblings aim at maximizing the sum of their survival rates, proved to be a key factor. We found that this principle is essential for the evolutionary stability of a population of cooperator homozygotes. Specifically, such population can never be stable in the anti-coordinated case, in which the cooperator-defector strategy pair yields the highest combined survival rate of siblings, regardless of the inheritance system (cf. Fig 11). In contrast, in the coordinated case in which the cooperator-cooperator

| Coordinated $2a_{11} > a_{12} + a_{21}$ | Recessive Cooperator | Dominant Cooperator | Intermediate |
|---|---|---|---|
| $A_1 = \begin{pmatrix} 0.6 & 0.1 \\ 0.7 & 0.4 \end{pmatrix}$ | Bistable | Bistable | Bistable |
| $A_2 = \begin{pmatrix} 0.55 & 0.1 \\ 0.6 & 0.3 \end{pmatrix}$ | Bistable | $G_1$ is ESGD $G_3$ is not ESGD | Bistable |
| $A_3 = \begin{pmatrix} 0.7 & 0.1 \\ 0.8 & 0.2 \end{pmatrix}$ | $G_1$ is ESGD $G_3$ is not ESGD | $G_1$ is ESGD $G_3$ is not ESGD | $G_1$ is ESGD $G_3$ is not ESGD |
| $A_4 = \begin{pmatrix} 0.5 & 0.1 \\ 0.8 & 0.15 \end{pmatrix}$ | $G_1$ is ESGD $G_3$ is not ESGD | Coexistence | Coexistence |
| $A_5 = \begin{pmatrix} 0.7 & 0.1 \\ 0.9 & 0.6 \end{pmatrix}$ | $G_1$ is not ESGD $G_3$ is ESGD | Bistable | Bistable |
| $A_6 = \begin{pmatrix} 0.6 & 0.3 \\ 0.8 & 0.5 \end{pmatrix}$ | $G_1$ is not ESGD $G_3$ is ESGD | $G_1$ is not ESGD $G_3$ is ESGD | $G_1$ is not ESGD $G_3$ is ESGD |
| $A_7 = \begin{pmatrix} 0.3 & 0.1 \\ 0.49 & 0.21 \end{pmatrix}$ | Coexistence | $G_1$ is not ESGD $G_3$ is ESGD | Coexistence |
| $A_8 = \begin{pmatrix} 0.6 & 0.3 \\ 0.8 & 0.44 \end{pmatrix}$ | Coexistence | Coexistence | Coexistence |

**Fig 10. Summary of prisoner's dilemma games in which the sum of the payoffs is maximized by the cooperator-cooperator strategy pair (coordinated case).** In the left column, the payoff matrices are given. The other columns contain the phase portraits of the corresponding genotype dynamics (1). Black dots indicate asymptotically stable rest points, while empty dots represent unstable equilibrium points. The bottom-left vertex of the triangle corresponds to the state (1,0,0), in which all individuals are ([a],[a]) cooperator homozygotes. The bottom-right vertex represents the state

(0,1,0), in which all individuals are heterozygotes with genotype ([a],[A]). The top vertex corresponds to the state (0,0,1), where all individuals are defector homozygotes with genotype ([A],[A]).

**Fig 11. Summary of prisoner's dilemma games in which the sum of the payoffs is maximized by the cooperator-defector strategy pair (anti-coordinated case).** See the caption at Fig 10.

strategy pair maximizes the sum of the siblings' survival rates, evolutionary stability of the cooperator genotype is possible (e.g., $A_3$ in Fig 10) but not guaranteed. Other outcomes such as bistability (e.g., $A_1$) and coexistence (e.g., $A_8$), or even global stability of the population of defector homozygotes may also arise (e.g., $A_6$). Thus, while the principle of "family welfare" plays a crucial role, the evolutionary stability of the population of cooperator homozygotes does not depend solely on it.

It is well known that theories based on group selection typically require at least two components for the spread of cooperation [49]: a) the synergistic effect of the number of cooperators on the fitness of group members, and b) the non-random group formation process, including composition.

In the context of familial selection discussed in this article, the family naturally defines the group, and its composition is determined by the genetic system. However, no synergistic effects are assumed, highlighting that group formation alone may be sufficient for the evolutionary stability of the population of cooperative homozygotes.

It is worth noting that standard models of group selection typically focus on asexual populations (e.g., [50,51]), and therefore are generally not suitable, or only limitedly applicable, for studying effects that emerge from group formation grounded in genetic foundations.

## Our results from the viewpoint of kin selection

The donation game provides a link between the prisoner's dilemma and the classical Hamilton's rule (e.g., [5,46,47]). It belongs to the coordinated case and involves only two parameters: cost and benefit. We found that regardless of the

genotype-phenotype mapping, the genotype dynamics leads to the asymptotically stable fixed point representing a population composed entirely of cooperator homozygotes when the classical Hamilton's rule is satisfied.

However, in general prisoner's dilemma games which typically involve more than two parameters, the evolutionary outcome is more complex. A population of homozygous defectors may be the unique evolutionarily stable state (see $A_6$ in Fig 10), but bistability and stable coexistence may also occur. Moreover, even a change in the genotype-phenotype mapping alone can alter the stability of the population of cooperator homozygotes (see $A_4$ in Fig 10), despite the payoff matrix remaining unchanged and relatedness being 1/2. It is also noteworthy that relatedness is not always a necessary condition for the stability of a homozygous cooperator population. For instance, inequalities (12) and (14) yield stability conditions without the usual multiplicative factor of 2 that reflects the 1/2 relatedness typical of siblings in monogamous population.

From the perspective of inclusive fitness, the cooperative gene can enhance its evolutionary success by indirectly increasing the survival probability of siblings who have a 1/2 chance of carrying the same allele. This indirect effect also operates under familial selection. The consideration of inclusive fitness thus explains the spread of the cooperative trait in models of kin selection. In line with this reasoning, we asked whether there exists a mechanism or condition within the framework of the genotype dynamics that guarantees an increase in the frequency of the cooperative allele along every trajectory. This led to introduce the concept of Haldane monotonicity. Unlike the inclusive fitness approach, however, the genotype dynamics not only allow us to characterize when a population of particular homozygotes can be asymptotically stable, but also to analyse phenomena such as bistability and the stable coexistence of different genotypes.

Van Veelen made an effort to identify conditions under which the classical Hamilton's rule provides accurate predictions. Using replicator dynamics, he showed that in linear models, the classical Hamilton's rule holds [47]. While evolutionary matrix games are indeed linear models, our focus differs from Van Veelen's: we study the genotype dynamics within the framework of Haldane's familial selection. In this setting, the framework of the classical Hamilton's rule cannot be sufficient (mathematically speaking) for deciding evolutionary stability. While the classical rule relies on two parameters (cost and benefit) in addition to the relatedness, our model generally requires three parameters to formulate the inequalities that determine evolutionary stability, based on the entries of the payoff matrix.

In summary, some of our results align with those of detailed models of group selection and kin selection, for instance, in the case of the donation game. However, there are notable differences. One of the most important distinctions is that our model highlights the substantial impact of the genotype-phenotype mapping on the outcome of natural selection. This difference may stem from a key feature of our approach, which explicitly incorporates diploidy, whereas most detailed models of kin selection, as noted by Van Veelen et al. [5, p. 180], are based on haploid assumptions (see, e.g., [6]).

## Genotype dynamics integrates core principles of kin and the group selection theories in familial selection

As noted above, the "welfare of the family" (that is, the sum of the survival rates of siblings) is a crucial determinant of the selection outcome. For instance, in the anti-coordinated case, where the cooperator-defector strategy pair maximizes the sum of the siblings' payoffs, the population of homozygous cooperator can never be evolutionary stable (as the total payoff is higher when one of the interacting siblings defects). On the other hand, "family welfare" alone is not sufficient to predict the evolutionary stability of the population of homozygous cooperators: as shown by the payoff matrix $A_6$ in Fig 10, where the population always evolves toward the homozygous defector state, regardless of the genotype-phenotype mapping.

In the case of the donation game, we have shown that the classical Hamilton's rule correctly predicts the evolutionary stability of a population consisting only of cooperator homozygotes. However, this arises the question: what is the precise role of the multiplier 2 in the inequality $b > 2c$ [52,53]? According to the classical Hamilton's rule, it reflects genetic relatedness. From the perspective of group selection, it corresponds to the number of players in the matrix game. In the context of the donation game, both interpretations appear reasonable.

In a diploid population, when two siblings interact with the aim of maximizing the "welfare of the family", both direct and indirect fitness effects come into play. In other words, under familial selection, the mechanisms of both kin selection and group selection are simultaneously at work. Thus, it can be concluded that the present model of familial selection, which focuses only on changes in genotype frequencies, bridges the foundational ideas of both theories within the framework of an "orthodox" Darwinian view.

## Conclusion

The genotype dynamics directly tracks changes in genotype frequencies over time. Unlike approaches based on the Price equation [54], individual-based or inclusive fitness [55], or the welfare of families [50], it operates without relying on these frameworks. Instead, the genotype dynamics captures the Darwinian struggle for existence, where the reproductive success of parents and the survival of offspring jointly determine the evolutionary success of each genotype. Although this dynamics is formally analogous to the well-known replicator dynamics, it belongs firmly to the tradition of classical population genetics. The genotypic composition within each family is determined by the details of the genetic system, while the genotype-phenotype mapping determines the phenotypic structure of the population.

One of our main findings is that the mathematical tools developed for classical asexual matrix games [41] can be successfully adapted to diploid population genetic models. In particular, the static definition of the evolutionarily stable genotype distribution (ESGD) implies the local stability of the corresponding state in the associated genotype dynamics (cf. [56,57], Theorem 7.2.4 in [41]). Furthermore, the genotype dynamics implicitly incorporates genetic systems where the genotype-phenotype mapping is well-defined, and the parental genotypes unambiguously determine the genotypes of their offspring. However, due to the potentially large number of genotypes, the resulting equations are often high-dimensional and non-linear. We hope that Theorem 1 presented here, along with the proposed static Definition 1 of evolutionary stability and the notion of Haldane monotonicity, will support both analytical and numerical studies of other selection regimes (see Remarks 1, 2 and 3) in future research.

In developing our applications, we started from Haldane's familial selection scenario [1], aiming to identify which phenotypes can survive in a monogamous family, assuming that the survival rates of siblings are determined by a game played among them. Our approach provides a direct and rigorous way to determine which behaviors are favored among siblings as a consequence of such interactions. It treats the population genetic aspects of the model with mathematical precision. In doing so, we follow the recommendation of Van Veelen et al. [58] to examine detailed assumptions explicitly. We focused on the simplest genetic system: a two-allele autosomal locus with Mendelian dominant-recessive or intermediate inheritance. Given an arbitrary payoff matrix, we identified the conditions under which a population of a specific homozygotes is evolutionarily stable and when an evolutionarily stable coexistence of individuals with different genotypes is possible.

As an application of our results, we examined the case where the interaction between siblings is modelled as a prisoner's dilemma. In this context, we provided concrete numerical examples (see Figs 10 and 11) that illustrate the conditions under which an evolutionarily stable mixed population composed of both cooperators and defectors can exist.

Our population genetic model, as shown by examples $A_2$, $A_4$, $A_5$, $A_7$, and $A_{10}$ in Figs 10 and 11, demonstrates that the genotype-phenotype mapping significantly influences the outcome of familial selection [3,29,36]. This finding is particularly interesting because, to our knowledge, neither kin selection nor multilevel selection models have previously captured this effect of the genotype-phenotype mapping.

In the vast majority of eusocial animals, such as the naked mole-rat, the family means the group. Similarly, in human societies, the family serves as the basic, traditional social unit [59]. In the light of this, we believe that familial selection provides a valuable framework for understanding the evolutionary origins of human altruism and cooperation. For example, altruistic behaviour plays a significant central role in the operation of family firms in the United States (e.g., [60]). As Bergstrom [61] emphasized, documented cases of familial altruism in human families may have evolutionary roots.

From the perspective of kin selection and group selection, our monogamous population genetic model of familial selection can be regarded as a special case in which the group is a genetically well-defined family and the survival of siblings is influenced solely by interactions among them. However, we emphasize that our model represents one of the simplest biological frameworks for investigating kin selection in diploid, sexually reproducing populations. Consequently, any comprehensive theory of kin selection should include our model as a special case.

Finally, we note that the genotype-centred perspective enables a fusion of classical asexual evolutionary game theory and sexual population genetics in the following sense. Mathematically, this perspective allows us to extend one of the most central results of asexual evolutionary game theory into the diploid population genetics. In a previous study [3], we directly applied the verbal definition of monomorphic evolutionary stability to genotypes in a diploid Mendelian population. In the present work, we introduced the definition of (mixed) evolutionarily stable genotype distribution (ESGD) for sexual diploid populations (see Definition 1). Building on this, and following the proof of the folk theorem of dynamical game theory (Theorem 7.2.4 in [41]) in the asexual setting, we proved that evolutionary stability of a genotype distribution implies stable coexistence of different genotypes (Theorem 1) under the genotype dynamics. We stress that Theorem 1 holds not only under pure familial selection but in a much more general context as well (see Remark 1, 2).

The genotype-centred approach offers several novel insights: First, it allows for the treatment of game theoretical conflicts between diploid genotypes under random mating. We showed that classifying matrix games in sexual diploid populations is a substantially more complex problem than in asexual ones. For example, while in classical asexual evolutionary game theory the prisoner's dilemma has a unique strict ESS (the defective phenotype), we found that when diploid full siblings play the prisoner's dilemma, a range of outcomes is possible: globally stable populations of either cooperators or defectors, globally stable coexistence or bistable situations (see Figs 10 and 11). In such cases, the outcome of the natural selection depends not only on the payoff matrix but also the genotype-phenotype mapping.

Second, within the genotype-centred framework accounting for the genotypes of mating individuals is indispensable. This task is greatly facilitated by employing mating tables. When diploid juveniles require parental care and grow up within families, they interact in ways that influence their survival and thus the selection process. In [37] we have already outlined several reasons why a genotype-centred population genetic model is essential for studying Haldane's familial selection regime. The main distinction between haploid kin selection models and Haldane's diploid familial selection model lies in that the latter explicitly incorporate the mating system [4], the genotype-phenotype mapping [3,4,37], and the linearity or non-linearity of the payoff function governing sibling interactions [4,37].

## Materials and methods

We introduced the general genotype dynamics (1) and the notion of evolutionarily stability for mixed diploid genotype ditributions. Following standard reasoning from classical game theory, we proved that evolutionary stability of a genotype distribution implies the local stability of the corresponding state in the genotype dynamics. Consequently, the static condition of evolutionary stability assist our dynamical analysis. Throughout our investigation, we employed standard mathematical analysis, game theoretical tools and Lyapunov's method. The figures were created using Wolfram Mathematica 12. The orbits shown in the phase portraits were obtained by numerically solving the differential equations with the NDSolve function in Wolfram Mathematica 12.

## Supporting information

**S1 File. Further mathematical details.**
(DOCX)

## Author contributions

**Conceptualization:** József Garay.

**Formal analysis:** Tamás Varga, Villő Csiszár, Tamás F. Móri.

**Investigation:** Tamás Varga, András Szilágyi.

**Methodology:** József Garay, Tamás F. Móri.

**Visualization:** Tamás Varga, András Szilágyi.

**Writing – review & editing:** József Garay, Tamás Varga, Villő Csiszár, Tamás F. Móri, András Szilágyi.

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
