## [Decision Letter · Decision Letter 0]

25 Sep 2024

Matrix game between full siblings in Mendelian populations

PLOS ONE

Dear Dr. Varga,

Thank you for submitting your manuscript to PLOS ONE. After careful consideration, we feel that it has merit but does not fully meet PLOS ONE’s publication criteria as it currently stands. Therefore, we invite you to submit a revised version of the manuscript that addresses the points raised during the review process.

We look forward to receiving your revised manuscript.

Kind regards,

Tao Fu

Academic Editor

PLOS ONE

Journal Requirements:

3. Thank you for stating the following financial disclosure: “This research was supported by the Hungarian National Research, Development and Innovation Office NKFIH (grant number 125569 received by TFM, grant number 140164 received by ASz). This research was supported by NKFIH, Hungary KKP 129877 received by TV. This research was supported by the Bolyai János Research Fellowship of the Hungarian Academy of Sciences received by ASz.” Please state what role the funders took in the study. If the funders had no role, please state: "The funders had no role in study design, data collection and analysis, decision to publish, or preparation of the manuscript." If this statement is not correct you must amend it as needed. Please include this amended Role of Funder statement in your cover letter; we will change the online submission form on your behalf.

4. We note that your Data Availability Statement is currently as follows: “All relevant data are within the manuscript and in Supporting Information files.”

Please confirm at this time whether or not your submission contains all raw data required to replicate the results of your study. Authors must share the “minimal data set” for their submission. PLOS defines the minimal data set to consist of the data required to replicate all study findings reported in the article, as well as related metadata and methods (https://journals.plos.org/plosone/s/data-availability#loc-minimal-data-set-definition). For example, authors should submit the following data: - The values behind the means, standard deviations and other measures reported; - The values used to build graphs; - The points extracted from images for analysis. Authors do not need to submit their entire data set if only a portion of the data was used in the reported study. If your submission does not contain these data, please either upload them as Supporting Information files or deposit them to a stable, public repository and provide us with the relevant URLs, DOIs, or accession numbers. For a list of recommended repositories, please see https://journals.plos.org/plosone/s/recommended-repositories. If there are ethical or legal restrictions on sharing a de-identified data set, please explain them in detail (e.g., data contain potentially sensitive information, data are owned by a third-party organization, etc.) and who has imposed them (e.g., an ethics committee). Please also provide contact information for a data access committee, ethics committee, or other institutional body to which data requests may be sent. If data are owned by a third party, please indicate how others may request data access.

Additional Editor Comments:

All reviewers have given positive feedback on the article and hope to make minor revisions in some details:

Reviewer 1:

Table 1 caption: Please remind the reader what n is.

Is there any intuition about the conditions 10 and 11? And, how do they relate to 2x2 games such as the hawk-dove, stag hunt, prisoner’s dilemma, and stag hunt? One may represent the qualitative space of 2x2 games in 2-dimensional space with parameters S and T (for an example see Santos 2006). The four quadrants are thus each a different game. It might be illustrative to represent the regions where the conditions 10 and 11 are met/not met on this plane. (Santos, Francisco C., Jorge M. Pacheco, and Tom Lenaerts. "Evolutionary dynamics of social dilemmas in structured heterogeneous populations." Proceedings of the National Academy of Sciences 103.9 (2006): 3490-3494.)

Tables 2 and 3: By “suspected” is it meant that a necessary but not sufficient condition is met?

Alternative states on Pg, 16 Line 407: Would not both defect in this case? If not, why not?

Reviewer 2:

I want it to be known that the authors did something sound and solid. And that does not mean that the paper cannot be improved. It can, but that concerns the presentation rather than there being something incorrect or conceptually wrong about their results. The paper does not really illustrate very well what we can learn from the replicator dynamics-like technique they developed (while I think it is great that they did). The paper covers what the authors call “the alternating case”, which adds to the overdose of cases and simplexes, and should really be an appendix, or maybe even a different paper. And the paper shies away a little bit from discussing how this relates to Hamilton’s rule in much detail. Given the state of the field, I do understand why they would avoid too much reflection on how Hamilton’s rule features in all of this, but given that we tend to use this example when we explain Hamilton’s rule, it would be better if the authors did do this.

In addition, please confirm again whether the article format is consistent with the requirements of this journal to avoid further modifications due to formatting issues in the future.

Reviewers' comments:

Reviewer's Responses to Questions

**Comments to the Author**

1. Is the manuscript technically sound, and do the data support the conclusions?

Reviewer #1: Yes

Reviewer #2: Yes

2. Has the statistical analysis been performed appropriately and rigorously?

Reviewer #1: N/A

Reviewer #2: N/A

3. Have the authors made all data underlying the findings in their manuscript fully available?

Reviewer #1: Yes

Reviewer #2: Yes

4. Is the manuscript presented in an intelligible fashion and written in standard English?

Reviewer #1: Yes

Reviewer #2: Yes

Reviewer #1: This paper develops a model of genotype dynamics, which combines Mendelian inheritance, replicator dynamics, and familial interactions. The model considers one autosomal locus with two alleles and thus three genotypes, which are mapped to phenotypes. There are then interactions between siblings that may be competitive or cooperative. These are represented by a 2x2 symmetric game where the phenotypes are strategies (either pure or mixed). Interactions occur between siblings, which creates a type of group selection dynamic. The paper develops a replicator-like equation for this system and presents the stability conditions (definitions and conditions) for this model, which are analogous to those in evolutionary game theory. In particular, it explores the conditions for stability of homozygous states and analyzes the scenario where the symmetric game is a Prisoner’s Dilemma. Several parsimonious conditions are found under which collaboration/cooperation is supported. Numerical solutions are used for more complicated scenarios. The paper is well written and thorough, and the model is interesting with many possible extensions as it folds genotypes and familial interactions into evolutionary game theory in a compelling and novel way. The results are well founded, and I found no errors in the calculations (the SI provides sufficient further details on calculations). In general, I enjoyed reading this paper and support its publication. I have only a few minor comments and questions detailed below.

Table 1 caption: Please remind the reader what n is.

Is there any intuition about the conditions 10 and 11? And, how do they relate to 2x2 games such as the hawk-dove, stag hunt, prisoner’s dilemma, and stag hunt? One may represent the qualitative space of 2x2 games in 2-dimensional space with parameters S and T (for an example see Santos 2006). The four quadrants are thus each a different game. It might be illustrative to represent the regions where the conditions 10 and 11 are met/not met on this plane.

Tables 2 and 3: By “suspected” is it meant that a necessary but not sufficient condition is met?

Alternative states on Pg, 16 Line 407: Would not both defect in this case? If not, why not?

Santos, Francisco C., Jorge M. Pacheco, and Tom Lenaerts. "Evolutionary dynamics of social dilemmas in structured heterogeneous populations." Proceedings of the National Academy of Sciences 103.9 (2006): 3490-3494.

Reviewer #2: I like this paper a lot. When we try to explain kin selection or Hamilton’s rule to students or to people outside academia, or when discuss it with each other, the go-to example is always full siblings – and for good reasons. Everyone in our profession has been doing this for about 60 years, and yet nobody has made a nuts-and-bolts, dynamic version of a model of a matrix game between full siblings. The authors however did exactly that, allowing for dominance and for deviations from equal gains from switching, and it is kind of unbelievable that they are the first. Whenever you think “why didn’t anyone do this before?” that tends to be a symptom that they did something that is very welcome to the field.

I have a bunch of points regarding the presentation (which I think is not very good) but it should be clear that I like what they do, and that they do it well. The reason why I would like to emphasize that, has to do with the following.

The field of inclusive fitness can be rather toxic and has some unscientific characteristics. If this paper falls in the hands of these people, I expect that their review reports will claim that this paper is fundamentally flawed (without a real indication why), that this paper is doing the same as they have already done in some other paper (without indicating exactly how), and preferably both (without seeing the irony in the combination). This is then typically reason enough for editors to reject the paper, even though both these claims might be completely untrue (but hard to check for the editor). So, I want it to be known that the authors did something sound and solid.

That does not mean that the paper cannot be improved. It can, but that concerns the presentation rather than there being something incorrect or conceptually wrong about their results. The paper does not really illustrate very well what we can learn from the replicator dynamics-like technique they developed (while I think it is great that they did). The paper covers what the authors call “the alternating case”, which adds to the overdose of cases and simplexes, and should really be an appendix, or maybe even a different paper. And the paper shies away a little bit from discussing how this relates to Hamilton’s rule in much detail. Given the state of the field, I do understand why they would avoid too much reflection on how Hamilton’s rule features in all of this, but given that we tend to use this example when we explain Hamilton’s rule, it would be better if the authors did do this.

I was planning to give a detailed description of how the presentation could be improved, and I am more than willing to do that still, but right now the deadline is approaching and I am running out of time. If the editors want to invite a resubmission, which I strongly suggest that they do, I am more than happy to make detailed suggestions how to improve the presentation of the paper, so that it makes it easy for the readers to understand why this is such a great contribution.

**Do you want your identity to be public for this peer review?** For information about this choice, including consent withdrawal, please see our Privacy Policy

Reviewer #1: No

Reviewer #2: No

---

## [Author Response · Author response to Decision Letter 1]

8 Nov 2024

See the attached file Response_to_Reviewers_GARAY_et_al_PlosONE.pdf

---

## [Decision Letter · Decision Letter 1]

17 Dec 2024

Dear Dr. Varga, 

Thank you for submitting your manuscript to PLOS ONE. After careful consideration, we feel that it has merit but does not fully meet PLOS ONE’s publication criteria as it currently stands. Therefore, we invite you to submit a revised version of the manuscript that addresses the points raised during the review process.

We look forward to receiving your revised manuscript.

Kind regards,

Tao Fu

Academic Editor

PLOS ONE

**Additional Editor Comments:**

In order to further improve the quality of this paper, a reviewer has proposed a series of modifications. The authors are asked to make further modifications.

Reviewers' comments:

Reviewer's Responses to Questions

**Comments to the Author**

Reviewer #1: All comments have been addressed

Reviewer #2: (No Response)

2. Is the manuscript technically sound, and do the data support the conclusions?

Reviewer #1: Yes

Reviewer #2: Yes

3. Has the statistical analysis been performed appropriately and rigorously?

Reviewer #1: N/A

Reviewer #2: N/A

4. Have the authors made all data underlying the findings in their manuscript fully available?

Reviewer #1: Yes

Reviewer #2: Yes

5. Is the manuscript presented in an intelligible fashion and written in standard English?

Reviewer #1: Yes

Reviewer #2: Yes

Reviewer #1: The authors have adequately responded to my comments and appropriately edited the manuscript. They also made several other cosmetic changes of which I approve.

Reviewer #2: Matrix game between full siblings in Mendelian populations - R1

By József Garay, Tamás Varga , Villő Csiszár, Tamás F. Móri, and András Szilágyi

Before I make a few broad suggestions how to improve the presentation, I want to emphasize how much I like the paper. This I would like to do to make sure that the authors understand that the critical remarks about the presentation are out of appreciation of what is in the paper, and that I am motivated by the wish that the paper is read, and the results are appreciated for what they are – and not disregarded or overlooked.

So, again, I would think that one of the selling points of this paper is that it does what should have been done long before, and that is not to take the shortcut of modeling interactions between siblings with an asexual model of kin selection; allow for relatedness to be r=1/2; and then think as interactions between siblings. Instead, this paper (also building on previous work by the same authors) does the homework right, and makes a proper model with sexual reproduction, with siblings that do get their genes from a father and a mother. Part of the improved presentation could, or should, be to stress this simple fact early on; point to the lack of a paper that models our prime example for kin selection properly, point to a few papers that take the asexual shortcut, and point out that this paper does not do that, with interesting differences. (Many papers also may not aim at doing what this paper does, but it is remarkable that we have a million papers for which asexual reproduction is the most natural interpretation, while everyone uses the sibling example to explain kin selection).

It seems that the paper focuses on identifying when we can expect one or the other behaviour to go to fixation; when we can expect bistability; and when we can expect coexistence. I think that this is a very good point to focus on, also if the paper wants to stress what having a proper model with sexual (rather than asexual) reproduction adds to our understanding of the dynamics. For that, it might help to stress a bit more that there are two relevant layers in the model; the payoff function and the genotype-phenotype mapping, both of which one could imagine contributing to this. This suggests that an interesting question to give a precise answer to here would be which of these two determines whether there is bistability or coexistence, or, more precisely, for which combinations of the two layers we get bistability or coexistence.

The introduction and abstract are a bit unspecific about this and they contain information that distracts from the main quest. The first sentence of the abstract, for instance, is way too opaque and technical, and only understandable by readers that are already familiar with the setup of the paper – which the authors obviously are, but most readers are not.

What also does not help is that the paper considers two scenarios. The first is called “cooperation” (collaborating case). The second is called “defector-cooperator strategy pair” (alternating case). First of all this is weird terminology, which I also do not really recognize from elsewhere. I would think “collaborating” is just a synomym for cooperation, and unless this is established terminology in some corner of the literature, this does not clarify anything. “Alternating” on the other hand is what a pair could do, in case it would repeatedly interact and have this payoff matrix, and the pair would want to get the largest combined fitnesses, and also distribute those in a balanced way. None of that happens in the paper, though, so why the word “alternating” is used, is unclear. (Also, this is a place where articles should be used; the collaborating/alternating case)

But, what is more important than the hurdle of the unusual and ambiguous terminology, is that for the second case it is super-obvious that neither can go to fixation. Therefore this case is an unneccessary distraction, especially given that, as of yet, in the introduction and abstract, it is not clear what makes bistability or coexistence possible in what the authors call the “collaborating” case. So I would suggest just to focus on the first case, with more precision, and let the second case be discussed in the appendix – which also allows the paper to not waste the readers attention in the abstract and intro on trying to figure out what these two cases are (most people will think of the first case when they hear the word prisoner’s dilemma anyway). Again, because I have seen the paper before, I know which payoff matrices go with the “collaborating” and the ”alternating” case, but at this point in the abstract, the unsuspecting reader can only guess, on the basis of a description that the authors surely think should be enough, but that really is not enough for readers to figure out what sets these two cases apart. Really explaining it there, on the other hand, would take too much time and space away from more interesting things that it could zero in on.

I do understand the wish for generality, and therefore the wish to cover all prisoners dilemmas. The wish to cover everything would be an argument for including the “alternating” case. However, it is also true that even if the “alternating” case is included, as it currently is, the paper still draws a line somewhere. It looks at prisoner’s dilemmas, and not at matrices for games that are not prisoner’s dilemma, that one could also study within the same framework. Therefore, I would think that one would have to ask what the added value is of drawing the line here (for the main text, and the main focus), where it includes the “alternating” case. I see more downsides than upsides of that. So, to be short, I would put the “alternating” case at in the appendix, to satisfy the desire for (more) completeness, but not have this as a hurdle to understanding how, for instance, prisoners’ dilemmas with or without equal gains from switching lead to different results, and how that interacts with the genotype-phenotype mapping. And, again, the terminology here should be improved.

I also want to respond to the author’s response to my earlier report. The authors write:

In any case, we note that we investigate a population genetic model with mathematical rigor. Our direct goal is not to examine Hamilton's original rule. The main reason for this is that Hamilton considered a "viscous" population, while we consider a population of monogamous families. Thus, it is not an obvious question how our results compare to Hamilton's original result.

In response to this, I would like to say two things. The first is that this contains an incorrect claim. A “viscous” poipulation is only one of the possible sources of relatedness suggested by Hamilton. As a matter of fact, this is an option that actually turned out to not work, or at least requires a deeper understanding of how kin selection works, if it does, to select altruistic/cooperative traits. Wilson, Dugatkin an Pollock (1992), and, subsequently, Taylor (1992), discovered the cancellation effect, when they were asking theirselves how viscosity could produce cooperation.

However, Hamilton (1964) does not model viscosity, or any other source for relatedness. Hamilton (1964) just postulated a vector of fitness effects, without specifying where this comes from.

Moreover, given that all of us use the example of siblings when explaining kin selection and Hamilton’s rule, I think it is important to relate this, not just to Hamilton’s original paper (however loosely), but also, and maybe more importantly so, to later ways in which Hamilton’s rule has been re-derived. This would also include models that one would naturally interpret as models with asexual reproduction, like Taylor (1989), to just give one example.

I am aware that this requires reorganizing and reworking the paper, but I think it will really pay off as for the dissemination of the results in this paper.

Smaller details

Throughout, articles are missing (for example; in donation game  in the/a donation game, but also in the title, where maybe the plural ). Also, the donation game in this setting is really a prisoners dilemma with equal gains from switching, as both siblings can choose.

Line 133: Is nk(ij) best described as a parameter? Later, it seems to be a function of the population state x.

References

Taylor, Peter D. "Evolutionary stability in one-parameter models under weak selection." Theoretical population biology 36.2 (1989): 125-143.

Taylor, Peter D. "Altruism in viscous populations—an inclusive fitness model." Evolutionary ecology 6 (1992): 352-356.

van Veelen, M., Allen, B., Hoffman, M., Simon, B., & Veller, C. (2017). Hamilton's rule. Journal of Theoretical Biology, 414, 176-230.

Wilson, David Sloan, Gregory B. Pollock, and Lee A. Dugatkin. "Can altruism evolve in purely viscous populations?." Evolutionary ecology 6 (1992): 331-341.

**Do you want your identity to be public for this peer review?** For information about this choice, including consent withdrawal, please see our Privacy Policy

Reviewer #1: No

Reviewer #2: No

---

## [Author Response · Author response to Decision Letter 2]

4 Feb 2025

See the attached file Response_2_to_Reviewers_GARAY_et_al_PlosONE

---

## [Decision Letter · Decision Letter 2]

22 May 2025

Thank you for submitting your manuscript to PLOS ONE. After careful consideration, we feel that it has merit but does not fully meet PLOS ONE’s publication criteria as it currently stands. Therefore, we invite you to submit a revised version of the manuscript that addresses the points raised during the review process.

We look forward to receiving your revised manuscript.

Kind regards,

Tao Fu

Academic Editor

PLOS ONE

Additional Editor Comments:

I admire the practical attitude of Reviewer 2 very much. I hope the authors can carefully study the opinions of Reviewer 2 and consider hiring native English speakers or professional English editing companies to help improve the English level of this paper.

Reviewers' comments:

Reviewer's Responses to Questions

**Comments to the Author**

Reviewer #1: All comments have been addressed

Reviewer #2: (No Response)

2. Is the manuscript technically sound, and do the data support the conclusions?

Reviewer #1: Yes

Reviewer #2: Yes

3. Has the statistical analysis been performed appropriately and rigorously?

Reviewer #1: N/A

Reviewer #2: N/A

4. Have the authors made all data underlying the findings in their manuscript fully available?

Reviewer #1: Yes

Reviewer #2: Yes

5. Is the manuscript presented in an intelligible fashion and written in standard English?

Reviewer #1: Yes

Reviewer #2: No

Reviewer #1: The reviewers have made a few additional changes as recommended by the other reviewer, which have helped improve clarity and should increase readership. This is a nice paper, and I'm pleased to recommend acceptance.

Reviewer #2: I copy paste from the attachment. Pleas look at the attachment, though.

Matrix game between full siblings in Mendelian populations - R1

By József Garay, Tamás Varga , Villő Csiszár, Tamás F. Móri, and András Szilágyi

I have expressed my appreciation for this paper, but I also find it a bit frustrating that the authors only pick up on suggestions very minimally, and incrementally. I find that frustrating, because I really want this paper to be published, but I also want people to be able to read it, understand it, and take it seriously. The authors also do not seem to be aware of their limitations in the English language and in writing in general. This puts me in a difficult situation; it is not my job to rewrite their paper, but the authors seem incapable or unwilling to present their results better.

To indicate this, let me just go over the title and abstract in some detail, as a sample.

Title: Matrix game between full siblings in Mendelian populations

This is not correct English. Correct options in are “Matrix games between full siblings in Mendelian populations” or “A matrix game between full siblings in Mendelian populations” or “The matrix game between full siblings in Mendelian populations”. To have a title in bad English is just embarrassing, and it should not be the case that the title alone already gives readers an excuse to ignore the paper. Out of the three correct options, the first would be the best, because the paper allows for a variety of 2x2 payoff matrices.

Abstract: The genotype dynamics pertain to Mendelian populations in which diploid individuals mate.

This is needlessly ambiguous. Which genotype dynamcs are we talking about here? Do you introduce them in this paper? Why not something like: we formulate/present/consider genotype dynamics in Medelian …?

It describes the temporal changes in the frequencies of the possible genotypes within a population of individuals following Mendelian inheritance rules.

This is an otherwise OK sentence, but why the “temporal”? Most changes are changes over time. This makes readers think they may have missed something.

We show that static evolutionary stability implies the stability of the corresponding interior equilibrium point in the genotype dynamics.

The static evolutionary stability is something that is also defined in this paper, right? Or is it an established definition?

We apply our findings to familial selection in a diploid, panmictic population, where the survival rates of siblings within monogamous and exogamous families are determined by a matrix game, and the behavior is uniquely determined by an autosomal recessive-dominant or intermediate allele pair. We provide conditions for the existence of each homozygote.

What do you mean “conditions for the existence of each homozygote”? Do you mean “conditions under which each homozygote is stable?”

In our numerical investigations of the prisoner’s dilemma between siblings, we distinguish between two cases depending on whether the sum of the survival rates of siblings involved in the interaction is greater in cooperator-cooperator interactions (cooperation case) or in cooperator-defector interactions (alternating case).

What are “numerical investigations”? You mean that you choose values for parameters and draw the dynamics on the simplex? Numerical sounds like it could be an alternative to analytic, which is not what any of this is about. Also, as before, I don’t like the terms “cooperation case” and “alternating case”. Whether or not to alternate is a choice one could make, not a property of a payoff matrix.

Based on the stability of the pure cooperator and defector states, we provide a potential classification of the genotype dynamics. We find that the pure cooperator population cannot fixate in the alternating case.

What does it mean for a cooperator population to fixate? You mean a mutant cooperator to go to fixation? Also, as I said before, this observation is trivial.

However, in the cooperation case, fixation is possible but not necessary, since bistability, coexistence, …

This is not a sentence. What does this mean? Totally vague.

… moreover, the monostable fixation of the pure defector state can also occur due to the interplay between the phenotypic payoff function and the genotype-phenotype mapping, which collectively determine the outcome of natural selection.

What does this mean? We are doing infinite population, deterministic dynamics, so why use the word fixation? Why not use terms like stable monomorphic state or something like that. And why leave the interplay a black box? These two ingredients are not both needed in conjunction to get bistability and/or coexistence.

In the donation game, the classical Hamilton’s rule implies the fixation of the cooperation in all considered genotype-phenotype mappings.

First of all “the cooperation” should be “cooperation”. Also, this is a strange way to put this. Hamilton’s rule does not imply fixation of cooperation in general, it is a rule that describes when cooperation does and when it does not fixate (again, if that is the teminology used, which I would argue against, because it really staes when cooperation is selected for, and with frequency dependent costs and/or benefits that can allow for stable mixtures).

And these are just superficial comments about the language. The abstract also does not really focus on what is most important.

I will leave it at this, but the rest of the paper has similar suboptimalities in the language.

Just to comment on something that the authors say in their response, in defense of not clarifying the usefulness of their approach in relation tot the inclusive fitness literature:

As mentiond, in the previous review, “The field of inclusive fitness can be rather toxic and has some unscientific characteristics.” Our previous experiences support this opinion. Therefore, whenever possible, we strictly focus on our population genetic model. If we start arguing in the Introduction in defense of our model—explaining why genotype-centered models are necessary for studying kin selection in diploid populations—we fear that we may lose some potential readers, particularly those who strongly adhere to Hamilton’s rule.

I do of course totally sympathise, but I think the risk-assesment by the authors is wrong. There are two aspects: 1) getting a paper published and 2) getting people to read it. Regarding point 1) you have two sympathetic reviewers, so there is no need to be overly cautious. Regarding point 2) the IF people will not read your paper anyway. Therefore, you should explain to neutral, non-partisan readers why this is helpful, and those are typically not offended by clarity and precision.

**Do you want your identity to be public for this peer review?** For information about this choice, including consent withdrawal, please see our Privacy Policy

Reviewer #1: No

Reviewer #2: No

---

## [Author Response · Author response to Decision Letter 3]

8 Jul 2025

See the attached document Response_3_to_Reviewers_GARAY_et_al_PlosONE.pdf

---

## [Decision Letter · Decision Letter 3]

11 Aug 2025

Matrix games between full siblings in Mendelian populations

PONE-D-24-15033R3

Dear Dr. Varga,

We’re pleased to inform you that your manuscript has been judged scientifically suitable for publication and will be formally accepted for publication once it meets all outstanding technical requirements.

Kind regards,

Tao Fu

Academic Editor

PLOS ONE

Additional Editor Comments (optional):

Reviewers' comments:

Reviewer's Responses to Questions

**Comments to the Author**

Reviewer #1: All comments have been addressed

Reviewer #2: All comments have been addressed

2. Is the manuscript technically sound, and do the data support the conclusions?

Reviewer #1: Yes

Reviewer #2: Yes

3. Has the statistical analysis been performed appropriately and rigorously?

Reviewer #1: N/A

Reviewer #2: N/A

4. Have the authors made all data underlying the findings in their manuscript fully available?

Reviewer #1: Yes

Reviewer #2: Yes

5. Is the manuscript presented in an intelligible fashion and written in standard English?

Reviewer #1: Yes

Reviewer #2: Yes

Reviewer #1: Though I approved the previous version, this revision has further improved clarity and precision. I recommend acceptance.

Reviewer #2: I am happy to see that the authors took the comments to heart, and I am looking froward to see the paper out. There may be some small differences in preferences remaining, but that is OK. I really appreciate this work; it is absolutely something useful for the field.

**Do you want your identity to be public for this peer review?** For information about this choice, including consent withdrawal, please see our Privacy Policy

Reviewer #1: No

Reviewer #2: No

---

## [Editor Report · Acceptance letter]

PONE-D-24-15033R3

PLOS ONE

Dear Dr. Varga,

I'm pleased to inform you that your manuscript has been deemed suitable for publication in PLOS ONE. Congratulations! Your manuscript is now being handed over to our production team.

Kind regards,

on behalf of

Dr. Tao Fu

Academic Editor

PLOS ONE